# Does wealth equate to happiness? an 11-year panel data analysis exploring socio-economic indicators and social media metrics

Feng Huang[1,2], Huimin Ding[3], Nuo Han[1,2,4], Fumeng Li[1,2], Tingshao Zhu[1,2]*

**1** CAS Key Laboratory of Behavioral Science, Institute of Psychology, Chinese Academy of Sciences, Beijing, China, **2** Department of Psychology, University of Chinese Academy of Sciences, Beijing, China, **3** School of Education, Renmin University of China, Beijing, China, **4** School of Data Science, City University of Hong Kong, Hong Kong SAR, China

* tszhu@psych.ac.cn

**Data Availability Statement:** The dataset and the code for the statistical analysis of this study are available on the Open Science Framework (OSF): https://doi.org/10.17605/OSF.IO/2TJZF.

## Abstract

The Easterlin paradox questions the link between economic growth and national well-being, emphasizing the necessity to explore the impact of economic elasticity, income inequality, and their temporal and spatial heterogeneity on subjective happiness. Despite the importance of these factors, few studies have examined them together, thus ongoing debates about the impact of economics on well-being persist. To fill this gap, our analysis utilizes 11 years of panel data from 31 provinces in China, integrating macroeconomic indicators and social media content to reassess the Easterlin paradox. We use GDP per capita and the Gini coefficient as proxies for economic growth and income inequality, respectively, to study their effects on the subjective well-being expressed by citizens on social media in mainland China. Our approach combines machine learning and fixed effects models to evaluate these relationships. Key findings include: (1) In temporal relationships, a 46.70% increase in GDP per capita implies a 0.38 increase in subjective well-being, while a 0.09 increase in the Gini coefficient means a 1.47 decrease in subjective well-being. (2) In spatial relationships, for every 46.70% increase in GDP per capita, subjective well-being rises by 0.51; however, this relationship is buffered by unfair distribution, and GDP per capita no longer significantly affects subjective well-being when the Gini index exceeds 0.609. This study makes a synthetic contribution to the debate on the Easterlin paradox, indicating that economic growth can enhance well-being if income inequality is kept below a certain level. Although these results are theoretically enlightening for the relationship between economics and national well-being globally, this study's sample comes from mainland China. Due to differences in cultural, economic, and political factors, further research is suggested to explore these dynamics globally.

## Introduction

The nexus between economic status and well-being has captivated thinkers from ancient Greece to the present. Solon argued that material wealth does not guarantee happiness [1], a

**Funding:** Tingshao Zhu received the Scientific Foundation of Institute of Psychology, Chinese Academy of Sciences (No.E2CX4735YZ). The sponsor had no further role in study design, in the collection, nalysis and interpretation of data, in the writing of the report, and in the ecision to submit the paper for publication.

**Competing interests:** The authors have declared that no competing interests exist.

sentiment echoed by Aristotle, who suggested that well-being could be attained through modest means [2]. This dialogue between ancient philosophy and contemporary empirical research highlights the timeless relevance of exploring the dynamics of economic growth and subjective well-being (SWB). SWB refers to an individual's self-assessment of their own happiness and life satisfaction, providing a comprehensive measure of personal and societal prosperity [3–5]. Notably, a seminal cross-cultural study revealed a positive correlation between economic growth and SWB in cross-sectional data, yet identified no consistent relationship in longitudinal analyses [6]. Termed the "Easterlin Paradox," this phenomenon underscores the complex interplay between economic advancement and SWB, a finding reinforced by a wealth of subsequent research [7–9].

Nonetheless, the association between economic growth and SWB has not gone unchallenged, especially when considering contemporary societal and policy issues such as debates around economic growth, environmental sustainability, and well-being. Evidence from several studies suggests that SWB correlates significantly with the material standard of living, highlighting the importance of integrating these findings into policy discussions to address the multifaceted dimensions of well-being beyond mere economic indicators [10,11]. Stevenson and Wolfers contended that there is no empirical evidence to suggest that a country's SWB plateaus after reaching a certain economic threshold, a point that gains relevance in the context of ongoing debates about the sustainability of economic growth and its implications for societal well-being [12]. Broadly speaking, Easterlin's findings enjoy robust support in many developed nations such as Western Europe, Japan, and Korea, serving as a critical reference point for policymakers aiming to balance economic growth with the enhancement of societal well-being [9,13,14]. However, results from some developing countries exhibit greater variability, suggesting that the path to well-being through economic growth may differ significantly across contexts, further underscoring the need for nuanced policy approaches that consider local socio-economic dynamics [12,15–17].

In the Chinese context, there exists a divergence of opinion regarding the Easterlin Paradox. One study focusing on six provincial capitals in China concluded that residents' happiness did not rise in tandem with national income [14]. Conversely, an analysis of the China General Social Survey (CGSS) from 2003 to 2010 found a consistent and statistically significant positive correlation between national income and SWB [18]. More recently, research efforts have shifted towards assessing the causal impact of economic growth on national happiness. Easterlin's research indicated that despite a fourfold increase in China's per capita GDP over the past two decades, SWB levels remained stagnant [19]. In contrast, other meta-analyses have revealed that the rising per capita GDP in China has exerted a positive influence on residents' SWB [20,21].

China serves as a particularly compelling case for analyzing the Easterlin Paradox, marked by its unmatched economic growth and substantial socio-economic shifts over recent decades. Unlike the steady and gradual economic progress seen in many Western countries, China has witnessed rapid industrialization, urbanization, and economic development at an unprecedented rate and scale. This transformation is characterized not only by its temporal aspect but also by significant economic disparities and variations in income distribution across and within its provinces. Our research aims to explore how these swift economic changes and notable shifts in income distribution impact individuals' perceptions of happiness. Focusing on China, we intend to provide insights that critically assess and refine the understanding of the Easterlin Paradox globally, promoting a more detailed and context-sensitive analysis.

To elucidate these disparities, our study examines the intricate relationships between economic growth, income inequality, and SWB in China. We suggest that the discrepancies noted in various studies could stem from the influences of economic elasticity, income inequality,

and their temporal and spatial heterogeneity on subjective happiness. Through the integration of psycholinguistic analysis of social media posts with economic indicators, such as per capita Gross Domestic Product (PRGDP) and the Gini coefficient, we aim to dissect the complex interactions among these factors and their effects on well-being. This methodology enables us to offer a more nuanced understanding of how economic factors influence subjective well-being, thus contributing to the ongoing discourse on the Easterlin Paradox and its relevance across different socio-economic environments.

Several scholars have explored the nexus between economic growth and SWB through the lenses of Hedonic Adaptation Theory [22] and Relative Deprivation Theory [23,24]. Extant literature suggests that individuals possessing elevated social status harbor greater anticipations for their future [25,26]. Unfulfilled mental expectations tend to impede the augmentation of SWB [27]. Moreover, happiness is fundamentally a relative construct [6], implying that an across-the-board elevation in income does not necessarily translate into a concomitant rise in overall SWB [28,29]. Although these theoretical frameworks are widely endorsed [25,27,28,30,31], comprehensive empirical examinations validating them remain conspicuously absent.

The current study advocates for a reevaluation of the Easterlin paradox through a theoretical lens. As posited by Hedonic Adaptation Theory, individuals within higher income brackets necessitate a more substantial increment in wealth compared to their lower-income counterparts to achieve a comparable elevation in happiness through income growth [32]. This disparity arises owing to their elevated baseline wealth and elevated mental expectations [33]. Such an observation suggests that the relationship between economic growth and SWB may inherently be nonlinear. Deaton (7)posited a semi-elastic effect on SWB exerted by economic growth [34], wherein changes in SWB correspond to percentage variations in regional economic expansion. Notably, the majority of extant studies have neglected this semi-elastic dimension, potentially contributing to the observed inconsistencies. We hypothesize that the relationship between regional economic growth and SWB is persistently positive across both cross-sectional and temporal dimensions when accounting for this elasticity. Specifically, the hypothesis can be articulated as follows:

**H1**: *Economic growth serves as a positive predictor of SWB in semi-elastic models, applicable to both cross-sectional and time-series analyses.*

Moreover, we advocate for the inclusion of local income inequality as a potential detrimental factor for SWB [35]. In accordance with Relative Deprivation Theory [29], income disparity engenders an escalation in relative deprivation among individuals, consequently undermining their well-being. Empirical evidence suggests that even middle-income residents experience diminished happiness due to social comparison when distributional inequality intensifies within a country or region, independent of changes in their real income. We posit that this adverse impact is notably pronounced in the short term, specifically in time-series analyses. This is informed by Hedonic Adaptation Theory, which posits that people may gradually acclimate to existing income disparities, redirecting their focus toward other life domains such as interpersonal relationships, health, and personal development. Hence, the deleterious influence of income inequality on SWB may be attenuated in long-term analyses, particularly in cross-sectional correlations, where economic factors assume greater significance in determining happiness. Accordingly, we formulate the subsequent hypothesis:

**H2**: *Income inequality exerts a negative influence on SWB within the temporal dimension.*

Although the direct deleterious influence of income inequality on SWB may be mitigated in cross-sectional analyses, it can still serve as a moderating factor that attenuates the positive

correlation between economic growth and SWB within the same framework. One rationale for this is that the principal contribution of economic growth to SWB lies in its capacity to elevate the quality of life for the broader populace. Nevertheless, a pronounced disparity between the wealthy and the impoverished within a specific region suggests that a minuscule fraction of the population monopolizes the majority of social resources [36]. This substantial wealth gap insinuates that the economic status of a region does not accurately reflect the living standards of its majority. In light of this, the current study formulates the subsequent hypothesis:

**H3**: *At the regional level, income inequality attenuates the positive predictive impact of economic growth on SWB.*

Our research employs these theories to construct a theoretical framework that elucidates the intricate relationship between economic growth, income inequality, and SWB (Subjective Well-Being). Specifically, we argue that while economic growth may initially enhance SWB, Hedonic Adaptation Theory suggests that this happiness tends to revert to a baseline level, thereby diminishing the linear impact of income increases. Concurrently, Relative Deprivation Theory explains how income inequality erodes well-being through social comparisons and undermines the positive effects of economic growth on SWB from a temporal perspective. Therefore, these theories form the basis of our hypotheses that economic growth exerts a semi-elastic positive impact on SWB across both spatial and temporal dimensions, and that income inequality not only directly reduces SWB but also moderates the positive influence of economic growth on SWB.

In response to the inconsistencies identified in previous research on the relationship between economic growth and SWB [37], our study aims to offer a more refined analysis by addressing the limitations of traditional analytical frameworks and incorporating considerations of elasticity. We utilize panel data to provide a comprehensive examination of the effects of macroeconomic indicators on SWB, a method that surpasses the constraints of purely cross-sectional or time-series analyses. Panel data models enable us to generate more robust estimates and examine the regional and temporal effects of economic variables on SWB within a cohesive dataset, thus enhancing the precision of our spatiotemporal analysis and mitigating inconsistencies due to random errors.

A significant innovation in our methodology is the application of a machine-learning model to analyze large-scale panel data, specifically focusing on the dynamic interaction between economic growth, income inequality, and SWB. This approach is particularly novel due to its use of social media big data as a proxy for continuous SWB measurement across diverse regions and time frames. Traditional psychological methods, such as questionnaires, are impractical for the extensive sampling required across China's thirty-plus provinces over a decade. In contrast, social media platforms like Sina Weibo, with its 566 million monthly active users and a dataset extending back to 2010 [38,39], provide a rich source for non-invasive public psychology and behavior analysis through big data analytics of user activities [40–42]. By leveraging this extensive dataset, our study employs a validated machine-learning model [4,43,44] to assess SWB based on the text of social media posts. This innovative approach allows us to explore the effects of economic growth and income inequality on SWB in both temporal and cross-sectional dimensions. Our findings aim to contribute a fresh perspective on the Easterlin paradox, particularly in the context of China's unique socio-economic landscape.

## Materials and methods

In alignment with extant research, this study employs PRGDP and the Gini coefficient as proxies for regional economic status and income inequality, respectively. Utilizing a validated

machine-learning model [4,43,44], we annually assess the SWB of each province over an 11-year span from 2010 to 2020. This timeframe is selected due to Sina Weibo's comprehensive user coverage beginning in 2010 and the availability of updated macroeconomic indicators from the China Statistics Bureau up to 2020. Fig 1 outlines the research methodology, encompassing data collection through to statistical analysis. Detailed descriptions of indicator selection, variable computation, and data processing can be found in the respective sections of the paper.

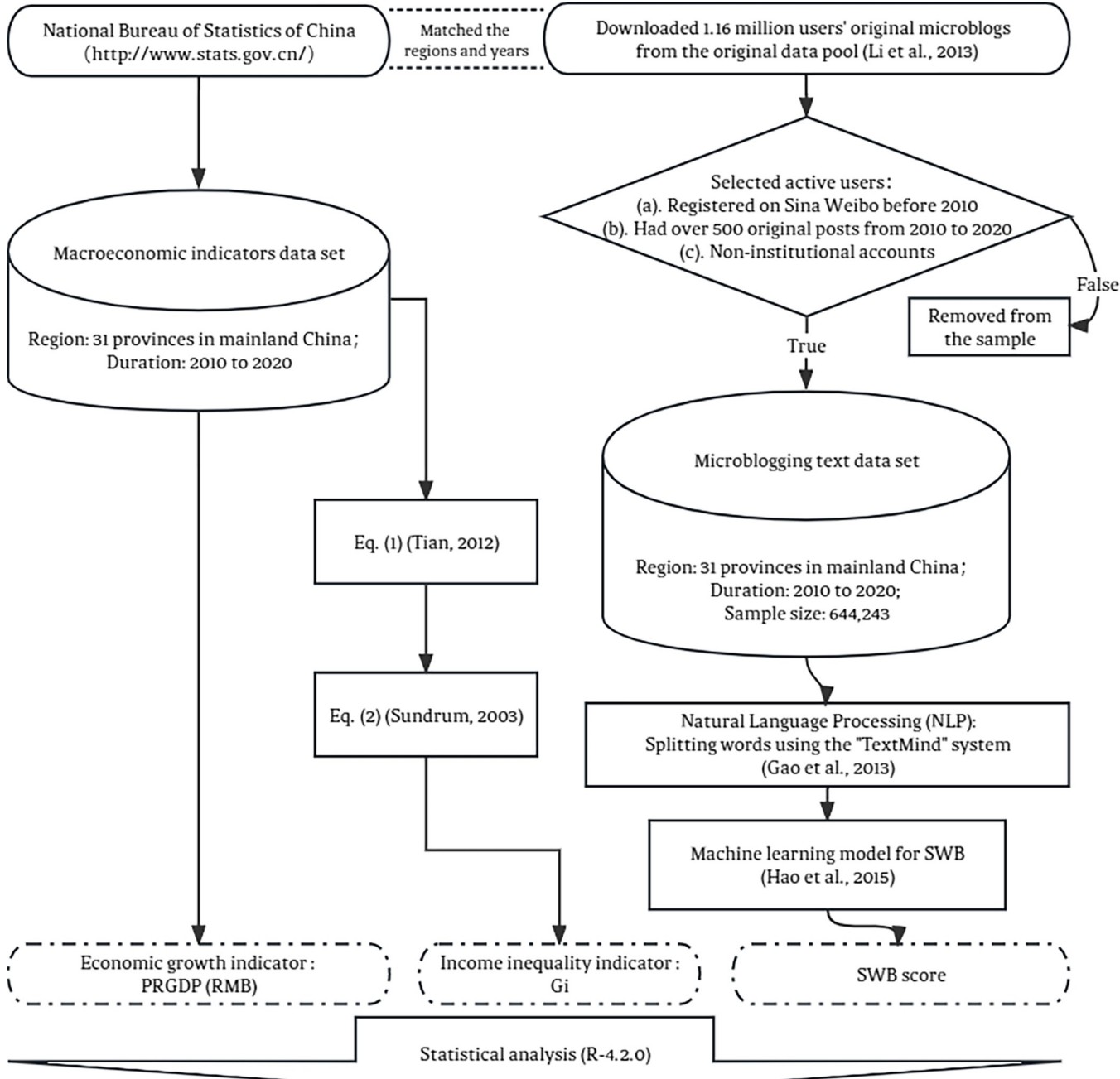

**Fig 1. The research process for the present study.**

## Data collection and preprocessing

The Application Programming Interface (API) serves as a software intermediary, enabling communication between two applications. In this case, Weibo's API facilitated systematic access to and collection of large-scale data from the platform, aiding in the analysis of user-generated content. For constructing regional Subjective Well-Being (SWB), text data was sourced from an initial pool of over 1.16 million Sina Weibo users [41]. Recognized as China's leading microblogging platform, Sina Weibo promotes the exchange and discussion of personal experiences, lifestyle activities, and celebrity news [45].

Following established data collection methods [42,44,46,47], we first retrieved public posts from 1.16 million mainland Chinese users via Weibo's API. Active users were then identified based on criteria including: (a) registration before January 1, 2010; (b) the exclusion of institutional, commercial, or celebrity accounts; and (c) a minimum of 500 original posts during the observation period. These criteria ensured the analysis focused on everyday shares from the general public, avoiding political or commercial propaganda, and aimed to include as broad a user base as possible. This led to the selection of 644,243 active Weibo users from thirty-one provinces for this study.

Upon completing data collection, the posts generated by the 644,243 active users were aggregated into thirty-one provinces across mainland China to align with corresponding macroeconomic indicators. To facilitate text analysis, we employed TextMind [48], a Chinese corpus processing software built upon the Linguistic Inquiry and Word Count (LIWC) framework [49,50]. This software segmented the text into individual words and identified psychosemantic features, enabling the subsequent calculation of province-specific SWB scores.

All macroeconomic variables utilized in this research are sourced from the National Bureau of Statistics of China (https://www.stats.gov.cn/). These include the PRGDP, employed as an indicator of each province's economic standing, as well as urban and rural population and income statistics, which are utilized to compute the Gini coefficient.

## Variable selection and calculation

**Subjective well-being.** This study conceptualizes SWB through Ryff's six-factor model [3], evaluating individuals' happiness by analyzing six psychological dimensions: self-acceptance, positive relationships with others, autonomy, environmental mastery, sense of purpose and meaning in life, and personal growth. This theoretical basis and measurement method receive high praise in the positive psychology arena [3–5]. By employing a validated machine learning model [4], we calculated a range of SWB metrics. The model uses a linear regression algorithm to determine the correlation between psycholinguistic attributes and self-reported scores, incorporating a 10-fold cross-validation method. Pearson correlation coefficients across all six SWB domains between predicted and self-reported scores were significantly strong [4,5]. Detailed discussions on the model's methodology and its applications are provided in [4,41,44]. Initially, the TextMind framework [48] was utilized to extract psycholinguistic features from the original posts of active Sina Weibo users. These features were then fed into the machine learning model to compute individual SWB metrics. Finally, an aggregate score was calculated by averaging these six metrics, consistently aligned with the original Likert scale, ranging from 1 to 7.

**Economic growth.** Aligned with the methodologies utilized in existing research [51,52], per capita Gross Domestic Product (PRGDP) is adopted as a metric for assessing the absolute economic growth of each province. PRGDP is defined as the quotient of the total GDP value and the regional population for a specified year. By adjusting for variations in city sizes, PRGDP provides a more accurate depiction of regional economic conditions compared to

GDP alone [42,53]. Simultaneously, the natural logarithm of PRGDP, ln(PRGDP), is used to gauge relative economic growth, facilitating the analysis of PRGDP's elasticity in relation to SWB—that is, the variation in residents' SWB in reaction to every doubling of the regional economy [34].

**Income disparity.** In accordance with established economic research paradigms, the Gini coefficient (Gi), first introduced by Corrado Gini [54,55], is utilized as the indicator to measure income inequality within each province [56]. This coefficient theoretically ranges from 0, indicating complete income equality, to 1, denoting absolute income inequality. Elevated values of the coefficient indicate higher levels of income disparity within a country or region. The computational methods applied in this context have received broad academic support for assessing income inequality across Chinese provinces, as detailed in Eq (1) [54] and Eq (2) [56].

$$Gi_a = 1 - \frac{1}{PW} \sum_{i=1}^{n} (W_{i-1} + W_i) \times P_i \tag{1}$$

In Eq (1), $Gi_a$ represents the Gini coefficient for income within either an urban or rural area of a given province; $P$ denotes the total population, $W$ signifies the aggregate income, and $W_i$ refers to the income accumulated by the $i^{th}$ group. Upon calculating the income-based Gini coefficients for both urban and rural populations in each province using Eq (1), the composite Gini coefficient for residents' income is subsequently computed through the group weighting method, as illustrated in Eq (2).

$$Gi = P_c^2 \frac{\mu_c}{\mu} G_c + P_r^2 \frac{\mu_r}{\mu} G_r + P_c P_r \frac{\mu_c - \mu_r}{\mu} \tag{2}$$

In Eq (2), $Gi$ denotes the provincial Gini coefficient for income; $G_c$ and $G_r$ specify the Gini coefficients for income among urban and rural residents, respectively. Furthermore, $P_c$ and $P_r$ indicate the population shares of urban and rural areas, while $\mu_c$, $\mu_r$ and $\mu$ represent the per capita income for urban, rural, and provincial populations, respectively.

## Statistical analysis

We constructed a panel dataset consisting of 31 provinces over an 11-year period to investigate the relationship between economic factors and subjective well-being (SWB) on both cross-sectional and time-series bases. Two variants of fixed-effects models based on least squares estimation were employed: individual fixed-effects estimation and time fixed-effects estimation. The former serves as a time-series regression with a linear constraint, wherein the regression coefficients capture the time-series correlations among the variables. Conversely, the latter functions as a cross-sectional regression with a linear constraint, and its coefficients reveal the cross-sectional correlations. Owing to the application of panel data and the removal of individual or time-related unobservables, the estimates from the fixed-effects models are deemed more robust than those generated by either cross-sectional or time-series regressions, as delineated in Eqs (3) and (4).

$$SWB_{i,t} = \lambda_i + \beta_1 PRGDP_{i,t} + \beta_2 Gi_{i,t} + \varepsilon_{i,t} \tag{3}$$

$$SWB_{i,t} = \delta_t + \beta_3 PRGDP_{i,t} + \beta_4 Gi_{i,t} + \varepsilon_{i,t} \tag{4}$$

Eq (3) represents an individual fixed-effects model, while Eq (4) illustrates a time-fixed effects model. In these equations, $SWB_{i,t}$ serves as the dependent variable, denoting the subjective well-being score for a specific province in a given year. Both *PRGDP* and *Gi* are

incorporated as independent predictor variables for SWB. $\lambda_i$ and $\delta_t$ account for the unobservable heterogeneity in province and year, respectively. $\varepsilon_{i,t}$ is designated as the error term.

In Eqs (3) and (4), the relationship between the variables is modeled as linear. Specifically, Eq (3) posits that, in the cross-sectional dimension, a one-unit increment in PRGDP corresponds to a $\beta_1$ alteration in SWB. Similarly, Eq (4) asserts that, in the time-series dimension, a one-unit increment in PRGDP yields a $\beta_3$ variation in SWB. Nevertheless, as highlighted in the introduction, we hypothesize that the influence of economic growth on SWB is nonlinear. This implies that the variation in SWB is contingent upon the percentage change in PRGDP rather than its absolute change, thereby leading us to consider the subsequent semi-elasticity models.

$$SWB_{i,t} = \lambda_i + \beta_1 \ln(PRGDP)_{i,t} + \beta_2 Gi_{i,t} + \varepsilon_{i,t} \qquad (5)$$

$$SWB_{i,t} = \delta_t + \beta_3 \ln(PRGDP)_{i,t} + \beta_4 Gi_{i,t} + \varepsilon_{i,t} \qquad (6)$$

The semi-elasticity models employed in this study adopt the logarithmic functional form commonly utilized in economic analyses [34]. Specifically, Eqs (5) and (6) build upon the linear models but substitute PRGDP with its natural logarithm. In the cross-sectional dimension, this implies that each doubling of PRGDP across regions results in a $\beta_1$ change in SWB. Conversely, in the time-series dimension, each doubling of PRGDP within a specific region induces a $\beta_3$ modification in SWB.

The current study formulated both linear and semi-elasticity models to assess their comparative efficacy in elucidating the relationship between economic indicators and SWB. To deepen this inquiry, an interaction term was constructed to investigate the synergistic effects of PRGDP and Gi on SWB. All statistical analyses were executed in R version 4.2.0, utilizing the plm package for fixed-effects modeling based on panel data [57], and the interactions package for evaluating moderating effects [58].

## Ethics statement

The research initiative received approval from the Ethics Committee, Institute of Psychology, Chinese Academy of Sciences (project number: H15009). All posts analyzed in this study were sourced from Sina Weibo, with stringent measures taken to preserve user privacy. The identities, usernames, and original content of the users' posts were deliberately omitted from the analysis, focusing exclusively on the examination of data at the provincial level. Given that Sina Weibo is publicly accessible and in adherence to the established protocols and ethical standards within the research domain [59], the necessity for informed consent was exempted by the Ethics Committee, Institute of Psychology, Chinese Academy of Sciences.

## Results

### Demographics

The demographic characteristics of the study participants are summarized in Table 1. Of the 644,243 active Sina Weibo users analyzed, 59.02% were female. A majority of users opted not to disclose their age; however, for those who did, ages ranged from 18 to 86 years, with the highest concentration falling within the 28- to 37-year age bracket, comprising 13.23% of the sample. Notably, 62% of the users hailed from Eastern China, a region generally regarded as more affluent compared to other parts of the country [46,60]. The provinces with the fewest participants had more than a thousand users each.

**Table 1. Demographic characteristics of selected participants.**

| Gender | male | 263,990 (40.977%) |
|---|---|---|
| | female | 380,253 (59.023%) |
| Age | 18–27 | 36,690 (5.695%) |
| | 28–37 | 852,36 (13.230%) |
| | 38–47 | 11,485 (1.783%) |
| | 48–57 | 1,976 (0.307%) |
| | 58- | 621 (0.096%) |
| | NA | 508,235 (78.889%) |
| Provinces | Center China | 111201 (17.261%) |
| | East China | 400642 (62.188%) |
| | West China | 132400 (20.551%) |
| Total | 644,243 (100%) | |

## Descriptive statistics

Table 2 presents the descriptive statistics for Subjective Well-Being (SWB), Per Capita Gross Domestic Product (PRGDP), and Gini coefficient (Gi) across 31 Chinese provinces spanning the years 2010 to 2020. Fig 2 illustrates the scatter plots and corresponding correlation matrices for these variables. Notably, a significant positive correlation exists among all macroeconomic indicators ($p < 0.001$), while no discernible relationship was observed between SWB and any of these indicators ($p > 0.05$). It is imperative to underscore that the correlation coefficients derived from panel data have circumscribed interpretive utility in the absence of constraints; thus, a subsequent fixed-effects analysis is requisite to elucidate the genuine relationships between these variables.

## Main effects estimation

Both the F-test and the Hausman test were conducted on the panel data models. The outcomes indicate that all F-tests yielded significant results ($p < 0.001$), suggesting the presence of unobservable individual or temporal heterogeneities. Moreover, the Robust Hausman test corroborates that these heterogeneities are associated with the independent variables ($p < 0.001$). Collectively, these tests advocate for the employment of fixed-effects models. The regression analyses involving PRGDP and Gi as predictors of SWB, across both time-series and cross-sectional dimensions, are summarized in Table 3.

As illustrated in Table 3, in the linear model for predicting SWB, the coefficient for Gi, as estimated through time-fixed effects, was not statistically significant ($p > 0.05$). In contrast, the coefficient for PRGDP was statistically significant ($B_{simple} = 0.23$, $p < 0.001$). When estimated by individual fixed effects, the coefficient for PRGDP was not significant ($p > 0.05$),

**Table 2. Descriptive statistics of variables.**

| Variable | Mean | SD | Median | Min | Max | Skewness | Kurtosis |
|---|---|---|---|---|---|---|---|
| SWB | 3.113 | 1.709 | 2.526 | 1.000 | 7.000 | 0.790 | −0.641 |
| PRGDP | 51219.944 | 26848.556 | 44348.000 | 12882.000 | 164158.000 | 1.666 | 3.259 |
| *ln* (PRGDP) | 10.730 | 0.467 | 10.700 | 9.464 | 12.009 | 0.316 | −0.026 |
| Gi | 0.277 | 0.085 | 0.267 | 0.077 | 0.551 | 0.589 | 0.745 |

Note. Cross-sections n = 31 (provinces), time points T = 11 (year), sample size N = 341; Abbreviation: SWB, Subjective well-being; PRGDP, GDP per capita (unit: RMB); ln (PRGDP), the natural logarithm of GDP per capita; Gi, Gini coefficient.

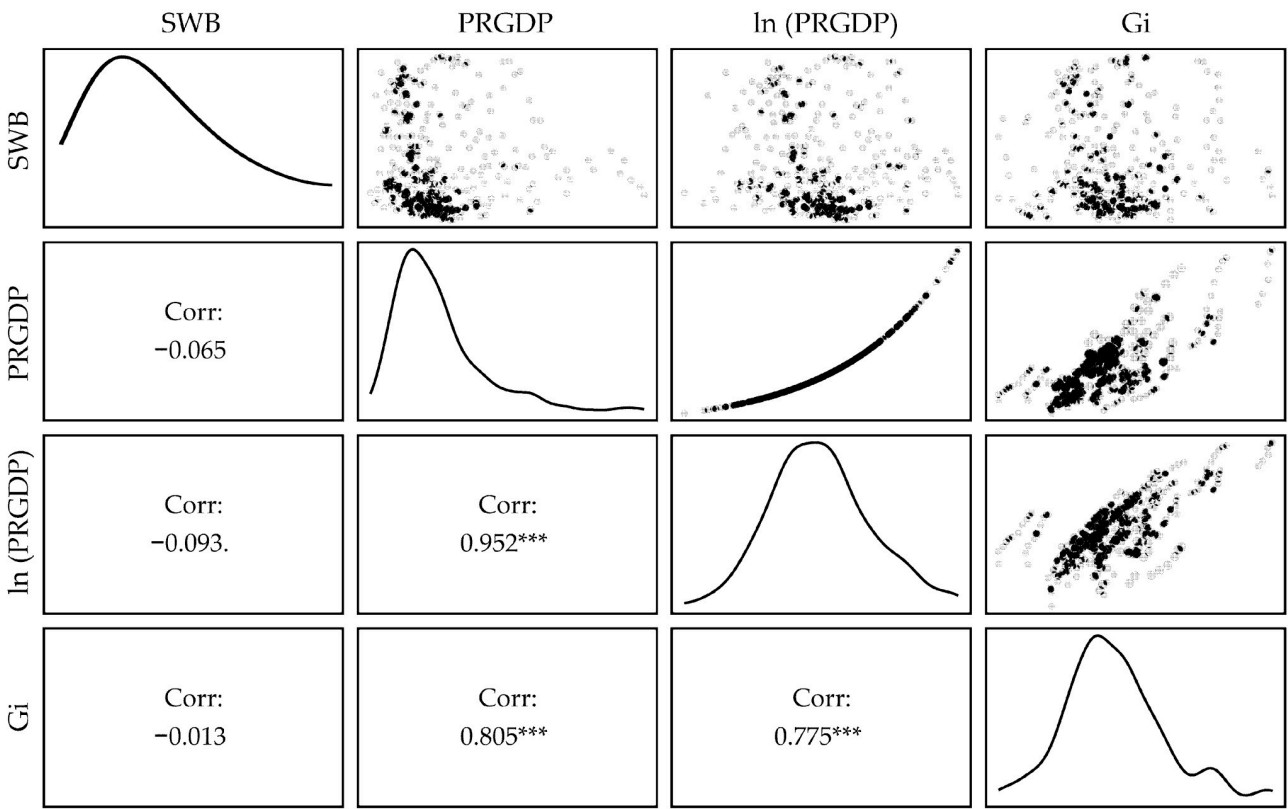

**Fig 2. The scattergram and correlation matrix for variables.**

**Table 3. The main effects of PRGDP and Gi on SWB.**

| Independent variables | Linear fit | | Elastic fit | |
|---|---|---|---|---|
| | **(1) Time-fixed** | **(2) Individual-fixed** | **(3) Time-fixed** | **(4) Individual-fixed** |
| PRGDP | 0.231*** | −0.075 | – | – |
| | (0.040) | (0.068) | – | – |
| *ln* (PRGDP) | – | – | 0.298*** | 0.216* |
| | – | – | (0.037) | (0.086) |
| Gi | 0.042 | −0.599*** | −0.005 | −0.864*** |
| | (0.040) | (0.068) | (0.037) | (0.086) |
| Adj. $R^2$ | 0.231 | 0.518 | 0.293 | 0.526 |
| F | 57.101*** | 198.526*** | 76.579*** | 204.305*** |
| Num. obs. | 341 | 341 | 341 | 341 |

Note. Standardized regression coefficients are displayed, with standard errors in parentheses. Abbreviation: SWB, Subjective well-being; PRGDP, GDP per capita (unit: RMB); ln (PRGDP), the natural logarithm of GDP per capita; Gi, Gini coefficient

\* $p < .05$.

\*\* $p < .01$.

\*\*\* $p < .001$.

whereas the coefficient for Gi was significant (Bsimple = -0.60, p < 0.001). In the semi-elasticity model, the coefficient for Gi, when estimated through time-fixed effects, was not significant (p > 0.05), but the coefficient for PRGDP was significant (Bsimple = 0.30, p < 0.001). Both the coefficients for PRGDP (Bsimple = 0.22, p < 0.05) and Gi (Bsimple = -0.86, p < 0.001) were statistically significant when estimated through individual fixed effects.

While all models converged successfully, the semi-elastic model exhibited superior fit compared to the linear model. In the time-fixed effects estimation, the semi-elastic model accounted for a greater proportion of the variance in SWB (29.30%) than did the linear model (23.10%). Similarly, in the individual fixed effects estimation, the semi-elastic model (52.60%) outperformed the linear model (51.80%). Collectively, these results suggest that the relationship between economic growth and SWB aligns more closely with an elasticity-based relationship.

According to the elasticity model's findings, in the time-series analysis, a 46.70% increase in per capita GDP (equivalent to 1 standard deviation) is associated with a 0.38 increase in the population's SWB (approximately 0.22 standard deviations). Conversely, a 0.09 increase in income inequality (equivalent to 1 standard deviation) leads to a decrement of 1.47 in well-being (roughly 0.86 standard deviations). In the cross-sectional analysis, a 0.09 increase in economic growth (equivalent to 1 standard deviation) elevates the population's overall happiness by 0.51 (approximately 0.30 standard deviations), while the change in SWB attributable to income inequality was not statistically significant.

## Interaction analysis

To explore the moderating role of Gi on the relationship between PRGDP and SWB, an interaction term for PRGDP and Gi was incorporated into the elasticity model. The fit indices are presented in Table 4.

As evidenced by Table 4, the interaction term proved significant in the time-fixed effects model (Bsimple = -0.08, p < 0.01), indicating that Gi does indeed moderate the cross-sectional relationship between PRGDP and SWB. According to the standardized coefficients, the positive impact of PRGDP on SWB diminishes by 0.14 (0.08 standard deviations) for each one-

**Table 4. The interaction of PRGDP and Gi on SWB.**

|  | (5) Time-fixed | (6) Individual-fixed |
|---|---|---|
| $ln$ (PRGDP) | 0.322*** | 0.186* |
|  | (0.038) | (0.093) |
| Gi | 0.018 | −0.837*** |
|  | (0.038) | (0.092) |
| $ln$ (PRGDP) × Gi | −0.077** | −0.031 |
|  | (0.029) | (0.036) |
| Adj. R2 | 0.307 | 0.525 |
| F | 54.445*** | 136.355*** |
| Num. obs. | 341 | 341 |

Note. Standardized regression coefficients are displayed, with standard errors in parentheses. Abbreviation: SWB, Subjective well-being; PRGDP, GDP per capita (unit: RMB); ln (PRGDP), the natural logarithm of GDP per capita; Gi, Gini coefficient.

* p < .05.

** p < .01.

*** p < .001.

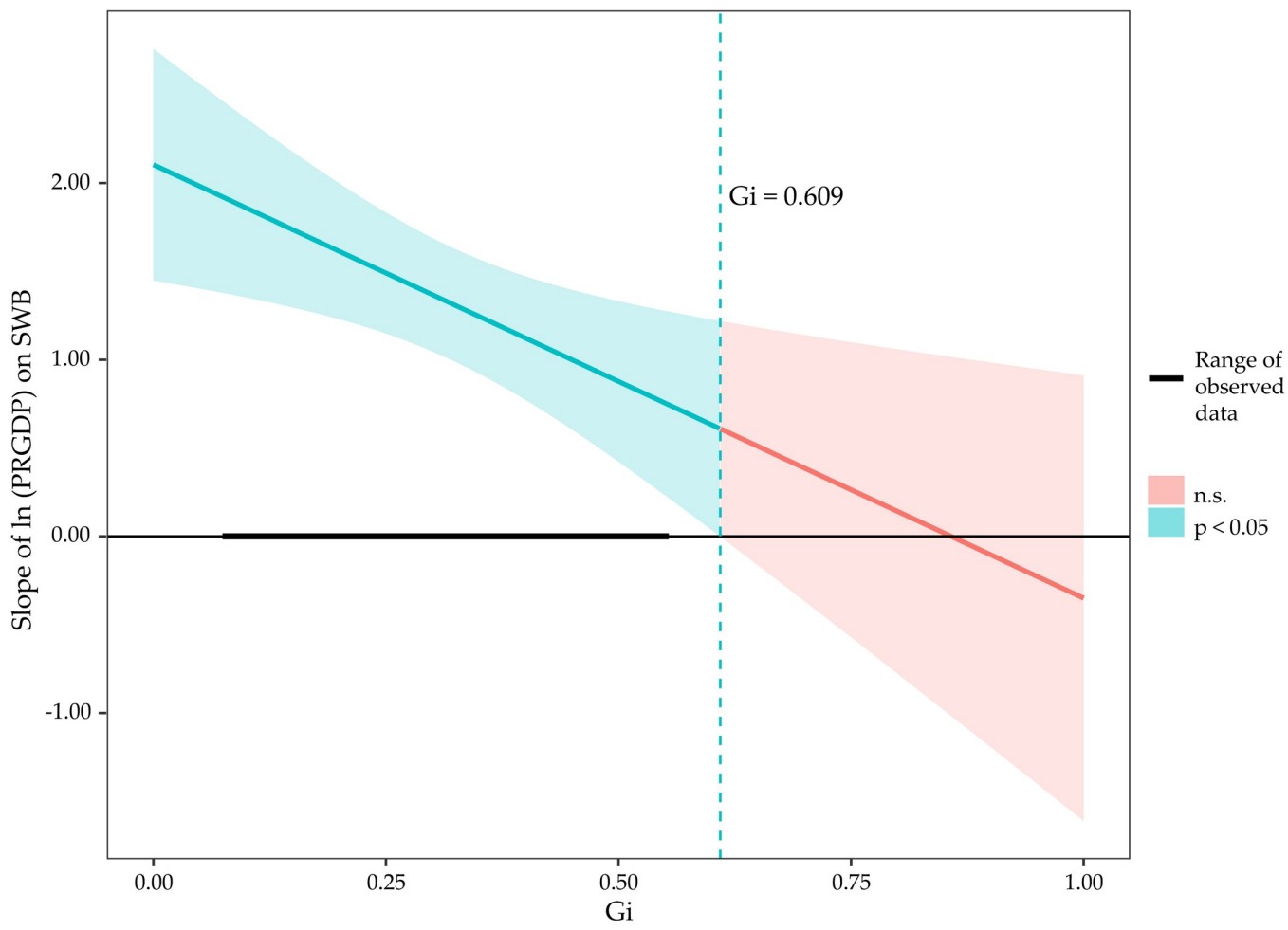

**Fig 3. The interaction of PRGDP and Gi on SWB (Johnson-Neyman analysis).**

standard-deviation increase in Gi. To further investigate this, Gi was categorized into high and low groups based on a mean value of ±1 standard deviation, and a simple slope analysis was conducted. The analysis revealed that PRGDP positively influences SWB in the low-Gi group (Bsimple = 0.446, $p < 0.001$), but its predictive power wanes in the high-Gi group (Bsimple declines from 0.446 to 0.332). For a more nuanced understanding of the interaction between PRGDP and Gi on SWB, we employed the Johnson-Neyman [61,62] technique to identify the critical value at which the regression coefficient becomes significant, as shown in Fig 3.

The findings indicate that the elasticity coefficient of PRGDP's positive influence on SWB diminishes with increasing Gi. Specifically, it is projected that when Gi surpasses the threshold of 0.609, the effect of PRGDP on SWB would become nullified, even though such a critical value of Gi does not exist within the range of our dataset.

## Discussion

David Hume, regarded as one of the pioneers of modern philosophy, asserted that the "The great end of all human industry is the attainment of happiness" [63,64]. Consistent with this view, subjective well-being serves as the cardinal aim for individual aspiration and for nations in steering their economic policies globally [65,66]. The current study probes into the interplay

between economic growth and income inequality as they pertain to subjective well-being, utilizing an extensive dataset of macroeconomic indicators and social media analytics across 31 provinces in China over an 11-year span. Our findings reveal bidirectional influences: a positive correlation exists between economic growth and subjective well-being, whereas income inequality exerts a negative impact in both temporal and cross-sectional contexts. Importantly, the enhancement of subjective well-being through economic growth is better described as elastic rather than linear. Furthermore, income inequality not only serves as a direct negative predictor but also attenuates the positive influence of economic growth on subjective well-being. These insights contribute to our understanding of the underlying dynamics of the Easterlin paradox and offer valuable guidelines for elevating societal well-being through economic advancements.

In the present study, we observe that the linear relationship between economic growth and subjective well-being in China aligns closely with the Easterlin paradox, which postulates that the positive cross-sectional correlation between these variables dissipates in time-series analysis [6]. However, when employing a more flexible model, we find a statistically significant positive correlation between the two variables across both time and cross-sectional dimensions, lending empirical support to our research hypothesis H1. These findings are in concordance with the hedonic adaptation theory's explanation of the Easterlin paradox, suggesting that the influence of wealth accumulation on happiness is not linear but is influenced by its baseline levels [22,27]. It is crucial to clarify that the observed elasticity in the relationship between economic growth and subjective well-being does not negate the claims of the Easterlin paradox. Rather, it indicates that as a region experiences exponential economic growth, there is a corresponding increase in the population's subjective well-being. Specifically, for every 46.70% increase in GDP per capita in the time series, subjective well-being rises by 0.38; in the cross-sectional analysis, a similar increase in GDP per capita yields a 0.51 rise in subjective well-being. Thus, economic factors remain pertinent, but the demand and expectation for growth have intensified.

Income inequality exerts a deleterious impact on subjective well-being, corroborating the tenets of relative deprivation theory [23,67]. In our study, for each 0.09 unit increase in income inequality in the time-series analysis, we observe a corresponding 1.47 unit decrease in subjective well-being, thereby confirming our Hypothesis H2 and aligning with extant literature. Luttmer posits that income inequality within a community skews the local mean income to the right of the median, representing a wealthier minority. Consequently, the majority of the population earns below the local mean, resulting in a decline in their 'relative income' [29]. Furthermore, our research illuminates a previously overlooked aspect: the interaction between income inequality and economic growth vis-à-vis subjective well-being. Specifically, the positive predictive influence of economic growth on subjective well-being diminishes as income inequality escalates. It is anticipated that the enhancement in well-being owing to economic growth vanishes entirely when the Gini coefficient attains a value of 0.61. This substantiates our Hypothesis H3. In accordance with relative deprivation theory, we surmise that the crux of this interaction lies in the widening wealth gap, which intensifies social comparisons and predominantly instills a sense of loss in the general populace [68,69].

The current findings underscore the elastic nature of economic growth in enhancing subjective well-being across both temporal and cross-sectional dimensions, in contrast to the adverse impact of income inequality. These observations align with prior research [7,10,12]. Conversely, other studies have posited a weaker association between economic growth and well-being [7–9,13,70]. We contend that these divergent findings may stem from an inadequate control for baseline economic conditions and income disparity. Consequently, we advocate for future research to rigorously address these potential biases when assessing the

influence of macroeconomic variables on mental well-being. Elasticity models should be employed to the extent possible, as they more accurately encapsulate the relationship between socio-economic factors and psychological well-being in both theoretical and empirical contexts. Simultaneously, the wealth disparity within specific locales should be scrutinized in tandem, as it may counteract the benefits of economic growth, thereby generating a misleading impression of non-correlation with subjective well-being.

Given our findings, the Easterlin paradox—which posits a lack of correlation between economic growth and subjective well-being in time-series analyses—is called into question. Specifically, in the context of China, we did not identify any temporal "threshold" beyond which economic growth ceases to contribute to subjective well-being. However, a discernible "threshold" for income inequality does exist. These insights underscore the premise that the enhancement of subjective well-being through economic growth is contingent upon a reasonably equitable distribution of social resources. Should the wealth gap surpass a critical juncture, the societal value of economic growth becomes largely moot for the majority of the population. These findings highlight the complex relationship between economic growth, income inequality, and subjective well-being, offering direction for policy development. To improve societal well-being, policies must foster sustainable economic growth and introduce initiatives to reduce income disparity. Recommended strategies encompass progressive taxation, enhancements in social welfare, policies promoting equal opportunities, and strengthening oversight mechanisms to deter tax evasion by affluent sectors and individuals. Such an approach to narrowing the wealth gap ensures that economic advancements lead to significant societal benefits. Thus, our study advocates for an integrated strategy in economic planning that not only seeks inclusive growth but also aims to boost overall happiness by ensuring equitable contributions across all economic sectors.

The current study employs social media big data and machine learning methodologies to explore the influence of macroeconomic variables on subjective well-being, thereby transcending some constraints inherent in previous disciplinary paradigms. Specifically, we introduce an innovative methodological framework that leverages real-time, rich data from social media platforms to elucidate the intricate and dynamic interplay between economic conditions and subjective well-being. This approach facilitates a nuanced analysis of macro-level determinants of subjective well-being, enabling the development of targeted policy interventions for distinct communities.

However, several limitations merit consideration. Firstly, our reliance on social media for data collection may exclude certain populations, such as individuals who are not literate or do not use social media. Over 78% of our sample from Sina Weibo did not disclose age data, constraining our ability to assess the impact of age demographics [71]. Future studies should employ varied sampling techniques to address this limitation. Secondly, as this study concentrates on the macro-social link between economic variables and collective subjective well-being, the findings might not be directly applicable to individual experiences, highlighting the potential for ecological fallacies. Future research could engage in individual-level investigations through surveys and experimental methods to draw comparisons with macro-level data. Thirdly, while our analysis focuses solely on the impact of objective economic indicators on subjective well-being, it does not delve into the psychological mechanisms underlying this relationship. Further studies are encouraged to explore mediating variables such as psychological expectations [72], social support [73], and cultural participation [74], in line with theories of hedonic adaptation and relative deprivation. Additionally, it is hypothesized that socio-cultural factors, including the emphasis on collective well-being, Confucian ethics' influence on notions of fairness and prosperity, and intra-national cultural differences, could significantly affect the interplay between economic growth, income inequality, and subjective well-being.

An in-depth examination of these elements might offer a more detailed perspective on how economic and social policies impact happiness in the Chinese context.

In conclusion, our study enriches the discourse on the intricate dynamics between economic growth, income inequality, and subjective well-being through an innovative analysis that combines macroeconomic indicators with social media analytics. By revealing the bidirectional influences where economic growth positively correlates with subjective well-being and income inequality negatively affects it, we offer nuanced insights into the debate surrounding the Easterlin paradox within the context of China's rapid economic evolution. Our findings challenge the traditional understanding of the paradox, demonstrating that economic growth can indeed enhance subjective well-being, provided income inequality is addressed effectively. This underscores the critical need for integrated economic policies that promote not only growth but also equity, to ensure that the benefits of economic advancements are widely shared across the society. Moreover, our methodological approach, leveraging social media data, opens new avenues for real-time, nuanced analysis of subjective well-being, offering a template for future research in the field. As we navigate the complexities of economic development and social welfare, the importance of fostering policies that minimize income disparities while bolstering economic growth has never been more evident. Our study ultimately advocates for a balanced approach to economic planning, one that equally prioritizes the pursuit of happiness and the equitable distribution of wealth, reflecting David Hume's assertion that the ultimate goal of human industry is the attainment of happiness.

## Conclusions

Utilizing social media big data and machine learning methodologies, this study investigates the macroeconomic determinants of subjective well-being. Our findings indicate the following: (1) In a time-series analysis, a 46.70% increase in GDP per capita corresponds to a 0.38 increase in subjective well-being, whereas a 0.09 uptick in the Gini coefficient leads to a 1.47 decline. (2) In cross-sectional data, a 46.70% surge in GDP per capita results in a 0.51 increment in subjective well-being, but a 0.09 rise in the Gini coefficient diminishes the positive predictive effect of GDP per capita on subjective well-being by 0.08. (3) According to the Johnson-Neyman technique, when the Gini index surpasses 0.609, GDP per capita ceases to be a significant contributor to subjective well-being. These insights suggest that economic growth can indeed foster national well-being, contingent upon maintaining income inequality below a critical threshold.

## Author Contributions

**Conceptualization:** Feng Huang, Huimin Ding, Fumeng Li, Tingshao Zhu.

**Data curation:** Feng Huang.

**Formal analysis:** Feng Huang.

**Funding acquisition:** Tingshao Zhu.

**Methodology:** Feng Huang, Huimin Ding, Nuo Han.

**Software:** Feng Huang.

**Supervision:** Tingshao Zhu.

**Validation:** Huimin Ding, Nuo Han.

**Visualization:** Feng Huang.

**Writing – original draft:** Feng Huang.

**Writing – review & editing:** Fumeng Li, Tingshao Zhu.

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
