## [Decision Letter · Decision Letter 0]

4 Jan 2024

PONE-D-23-36509Does Wealth Equate to Happiness? An 11-Year Panel Data Analysis Exploring Socio-Economic Indicators and Social Media MetricsPLOS ONE

Dear Dr. Zhu,

Thank you for submitting your manuscript to PLOS ONE. After careful consideration, we feel that it has merit but does not fully meet PLOS ONE’s publication criteria as it currently stands. Therefore, we invite you to submit a revised version of the manuscript that addresses the points raised during the review process.

We look forward to receiving your revised manuscript.

Kind regards,

Roghieh Nooripour, Ph.D

Academic Editor

PLOS ONE

Journal Requirements:

2. In your Methods section, please include additional information about your dataset and ensure that you have included a statement specifying whether the collection and analysis method complied with the terms and conditions for the source of the data.

4. In the online submission form, you indicated that To protect the participants’ privacy, the original posts used for the analysis are not publicly available but from the corresponding author at a reasonable request. The processed data set and R code are available on https://www.scidb.cn/anonymous/MnUyYXF1/. 

3. Uploaded as supplementary information.

Reviewers' comments:

Reviewer's Responses to Questions

**Comments to the Author**

1. Is the manuscript technically sound, and do the data support the conclusions?

Reviewer #1: Yes

Reviewer #2: Yes

2. Has the statistical analysis been performed appropriately and rigorously? 

Reviewer #1: I Don't Know

Reviewer #2: Yes

3. Have the authors made all data underlying the findings in their manuscript fully available?

Reviewer #1: Yes

Reviewer #2: Yes

4. Is the manuscript presented in an intelligible fashion and written in standard English?

Reviewer #1: Yes

Reviewer #2: Yes

5. Review Comments to the Author

Reviewer #1: Greetings and Regards,

Thank you for giving me the opportunity to read this valuable manuscript.

The title and abstract are appropriate and the entire text is written in a standard. In the following section, there are points that can help to improve the quality of the article and better understanding of the readers.

1. The opening sentences of an abstract make a big claim. Have all the articles in this field been reviewed and all the scientific articles and texts have ignored the elasticity of economic determinants on subjective well-being and the implications of income

inequality? It is suggested that this sentence be expressed less confidently or rewritten in another way.

2. For a better understanding of the readers, it is better to define the research variables in the introducion.

3. Considering that social media has been used to gather and analyze the data of the target sample, What will happen to the percentage of people who are not literate or do not use social media? These cases can be added to the limitations of the research.

4. It is suggested to explain about the TextMind framework so that the reader gets to know it better.

Reviewer #2: The topic is fascinating, but a thorough restructuring of the content is necessary. I kindly request the esteemed author to revise the entire article with a deeper and more coherent perspective, taking into consideration the suggested changes. Afterward, please resubmit it for further review. I believe this feedback will be valuable in improving the quality of your work.

Best regards.

Abstract

1. Highlight how your study addresses these gaps or adds new insights to the existing body of knowledge would strengthen the abstract.

2. The use of technical terms like the Johnson-Neyman technique might not be immediately clear to all readers. A brief explanation or simplification of these terms can make your abstract more accessible to a broader audience.

3. Ensure the economic and social indicators used in your study are clearly defined and relevant to your research question. Clarifying how these indicators specifically relate to subjective well-being in the Chinese context would be beneficial.

4. Since your study is focused on China, a brief mention of the extent to which these findings might be generalizable to other regions or economies would be useful.

Introduction

You begin with historical perspectives on well-being, which is a strong start. However, to maintain relevance, quickly transition to how these views relate to modern economic theories and findings.

provide a more detailed explanation for readers unfamiliar with the concept. This helps in setting a clear foundation for your study.

Deepen this section by briefly discussing why such divergences exist and what your study aims to contribute to this debate.

Clearly articulate why China is a particularly relevant or unique case for studying this paradox. This will help in establishing the significance of your research.

You mention Hedonic Adaptation Theory and Relative Deprivation Theory but could integrate these more cohesively into your argument. Explain how these theories underpin your hypotheses.

Offer a brief preview of your methodology, especially the use of a machine learning model and panel data. This sets the stage for what makes your approach novel or robust.

consider providing a bit more context or rationale for each. This can help readers understand how you arrived at these specific hypotheses.

Tie your study to contemporary societal or policy issues, such as the ongoing debates about economic growth and well-being. This can enhance the real-world relevance of your research.

Methods

1. Ensure that all technical terms, especially those specific to economics and machine learning, are clearly defined or explained. This aids in accessibility for readers who may not be familiar with these fields.

2. adding a bit more detail on how you addressed potential biases in social media data could be beneficial. This might include discussing the representativeness of the Sina Weibo user base or the limitations of social media data.

3. You’ve detailed how SWB and other variables were calculated. Explain why these particular variables were chosen and how they are expected to contribute to your research objectives would provide more context.

4. Elaborate on why this approach was chosen, its benefits, and any limitations it might have compared to traditional methods.

5. If there are any ethical considerations, especially related to the use of social media data, they should be explicitly stated. This might include how user privacy was protected or how data anonymization was handled.

Results

1. consider simplifying the presentation for better readability. Use clear subheadings to separate different aspects of the results, such as demographics, descriptive statistics, main effects estimation, and interaction analysis.

2. provide brief interpretations of what these statistics mean in terms of your research question. Discuss the practical implications of these findings, especially in the context of policy and economic planning.

3. Elaborate on the implications of the findings related to the Gini coefficient and PRGDP. How do these results contribute to existing literature, and what do they suggest for future economic and social policies?

Discussion

1. You've adeptly connected your findings to hedonic adaptation theory and relative deprivation theory. Further elaborating on how your results extend or challenge these theories could add depth to your discussion.

2. provide a clearer articulation of how these insights can inform policy-making or economic planning would be beneficial.

3. Delv deeper into how your findings align with or diverge from these studies could offer a richer context.

4. Expanding on how limitations might impact the interpretation of your results and suggesting ways future research could overcome them would strengthen your discussion.

5. Discuss potential applications of your findings beyond the Chinese context. How might these insights apply to other economies or inform global economic policies?

6. Reflect on the strengths and limitations of using social media data and machine learning methodologies in your research. How do these methods compare to traditional survey-based approaches in studying subjective well-being?

7. Suggest specific areas for future research that could build on your findings. For instance, exploring individual-level data or integrating psychological variables could provide further insights.

8. on your findings, offer specific recommendations for economic policies that could enhance subjective well-being while addressing income inequality.

9. Discuss any socio-cultural factors that might influence the relationship between economic growth, income inequality, and subjective well-being, particularly in the Chinese context.

10. End your discussion with strong concluding remarks that summarize the key takeaways of your study and their broader significance.

6. PLOS authors have the option to publish the peer review history of their article (what does this mean?). If published, this will include your full peer review and any attached files.

Reviewer #1: No

Reviewer #2: **Yes: **Roghieh Nooripour

---

## [Author Response · Author response to Decision Letter 0]

2 Mar 2024

Dear Reviewers,

Thank you for the comments concerning our manuscript entitled 'Does Wealth Equate to Happiness? An 11-Year Panel Data Analysis Exploring Socio-Economic Indicators and Social Media Metrics' (Ref.: PONE-D-23-36509). Those comments are valuable and very helpful for revising and improving our paper and the important guiding significance to our researchers. We have carefully considered the suggestions and made changes. Revised portions are highlighted by using the track changes mode in WORD. The responses to comments are as follows:

Response to Reviewer 1

1. The opening sentences of an abstract make a big claim. Have all the articles in this field been reviewed and all the scientific articles and texts have ignored the elasticity of economic determinants on subjective well-being and the implications of income inequality? It is suggested that this sentence be expressed less confidently or rewritten in another way.

Response 1: Thank you for your constructive feedback on the opening sentences of our abstract. We have taken your suggestion into account and revised the section to more accurately reflect the current state of research in this field. The modified content is as follows:

"... A critical focus is required on the elasticity of economic growth and the boundary conditions of income inequality, as well as their temporal and spatial heterogeneity in influencing subjective well-being. However, there are scarcely any studies that simultaneously consider these factors, leading to ongoing debates regarding the relationship between economic factors and well-being. ..."

We believe this adjustment improves the precision of our statement and sets a more accurate context for our research contributions.

2. For a better understanding of the readers, it is better to define the research variables in the introducion.

Response 2: Thank you for your insightful recommendation to define research variables in the introduction for enhanced reader comprehension. In the revised version of our manuscript, we have added brief definitions of all variables within the "Introduction" section, specifically in paragraphs detailing the theoretical background and rationale of our study. Furthermore, we provide detailed explanations of these variables in the subsequent "Methods" section to ensure a thorough understanding.

3. Considering that social media has been used to gather and analyze the data of the target sample, What will happen to the percentage of people who are not literate or do not use social media? These cases can be added to the limitations of the research.

Response 3: In response to your insightful comment, we have acknowledged this concern within the "Discussion" section of our manuscript, emphasizing the limitations associated with our data collection method. The pertinent excerpt is as follows:

“… Our reliance on social media for data collection may exclude certain populations, such as individuals who are not literate or do not use social media. Over 78% of our sample from Sina Weibo did not disclose age data, constraining our ability to assess the impact of age demographics. Future studies should employ varied sampling techniques to address this limitation.”

We have critically discussed its implications for our research's comprehensiveness and generalizability. This acknowledgment serves to underline the necessity for future studies to develop and implement strategies that encompass a broader spectrum of the population, thereby enriching the representativeness and applicability of subsequent findings. We appreciate your attention to this matter, which significantly contributes to the rigorous evaluation of our study's scope and limitations.

4. It is suggested to explain about the TextMind framework so that the reader gets to know it better.

Response 4: In response to your suggestion, we have expanded the manuscript to include a more comprehensive introduction to the TextMind framework. Specifically, we stated: 

"… We employed TextMind (Gao et al., 2013), a Chinese corpus processing software built upon the Linguistic Inquiry and Word Count (LIWC) framework (Tausczik & Pennebaker, 2009; Zhao et al., 2016). This software segments the text into individual words and identifies psychosemantic features, enabling the subsequent calculation of province-specific Subjective Well-Being (SWB) scores." 

We believe this addition will provide readers with a clearer understanding of the methodologies applied in our research, enhancing the transparency and replicability of our study. Thank you for highlighting the importance of elucidating the tools and frameworks utilized in our analysis.

Response to Reviewer 2

Abstract

1. Highlight how your study addresses these gaps or adds new insights to the existing body of knowledge would strengthen the abstract.

Response 1: In response to your recommendation, we have revised the abstract to more explicitly highlight how our study addresses existing gaps and contributes new insights to the body of knowledge. Your suggestion has been invaluable in enhancing the clarity and impact of our abstract, and we are grateful for your guidance in this improvement.

2. The use of technical terms like the Johnson-Neyman technique might not be immediately clear to all readers. A brief explanation or simplification of these terms can make your abstract more accessible to a broader audience.

Response 2: Acknowledging your concern regarding the accessibility of technical terms such as the Johnson-Neyman technique to all readers, we have modified our approach in the revised version of the abstract. To maintain the clarity and brevity of the abstract, we have omitted references to these non-central elements. Instead, we have opted to provide a detailed explanation of such terms in the methods section, where they are contextualized within our study's framework and supported by relevant citations. This adjustment ensures the abstract remains accessible to a broader audience while preserving the integrity and specificity of our methodological descriptions. We appreciate your guidance in making our research more inclusive and understandable for readers from diverse backgrounds.

3. Ensure the economic and social indicators used in your study are clearly defined and relevant to your research question. Clarifying how these indicators specifically relate to subjective well-being in the Chinese context would be beneficial.

Response 3: In consideration of your suggestion to ensure clarity regarding the economic and social indicators used in our study, we carefully deliberated on the necessity of defining and explaining these indicators, such as per capita GDP and income inequality, within the abstract. Given their widespread recognition and self-explanatory nature, along with the imperative to maintain the succinctness of the abstract, we opted not to elaborate on them in this section. Instead, we have enhanced the introduction by including a detailed discussion on how these indicators specifically relate to subjective well-being in the Chinese context. Additionally, we have provided a more comprehensive description of the involved variables in the methods section. This approach allows us to keep the abstract concise while ensuring that readers can find a thorough exposition of the indicators and their relevance to subjective well-being in China in the corresponding sections of the paper. We are grateful for your feedback, as it has guided us in striking a balance between conciseness and informativeness.

4. Since your study is focused on China, a brief mention of the extent to which these findings might be generalizable to other regions or economies would be useful.

Response to Comment 4: In response to your insightful suggestion, we have amended the abstract to incorporate a discussion on the generalizability of our findings to other regions or economies. Specifically, we have added the following statement at the end of the abstract: 

"… Although these results are theoretically enlightening for the relationship between economics and national well-being globally, this study's sample originates from mainland China. Due to variances in cultural, economic, and political factors, we recommend further research to explore these dynamics in a global context." 

We are grateful for your recommendation, as it enhances the completeness and scholarly utility of our work by addressing the potential for broader applicability and encouraging continued exploration in this vital area of research.

Introduction

1. You begin with historical perspectives on well-being, which is a strong start. However, to maintain relevance, quickly transition to how these views relate to modern economic theories and findings.

Response 1: In alignment with your valuable feedback, we have revised the introduction of our manuscript, particularly the opening section, to ensure a smoother transition from historical perspectives on well-being to their relevance within the context of modern economic theories and findings ("Introduction" section, first paragraph). We appreciate your guidance in refining our introduction to better capture the interplay between historical and modern perspectives on well-being.

2. provide a more detailed explanation for readers unfamiliar with the concept. This helps in setting a clear foundation for your study.

Response 2: Taking into account your constructive feedback, we have made significant improvements in our revised introduction to ensure a clearer and more detailed explanation of key concepts, such as the "Easterlin Paradox." We believe that this enhancement will facilitate a deeper engagement with our work and appreciate your suggestion as it has significantly contributed to the refinement of our manuscript.

3. Deepen this section by briefly discussing why such divergences exist and what your study aims to contribute to this debate.

Response 3: In response to your insightful suggestion, we have delved into a deeper discussion based on the Hedonic Adaptation Theory and Relative Deprivation Theory in the revised manuscript, aiming to hypothesize the divergences found in previous research (“Introduction” section, paragraphs 7-9). This discussion is directly related to the formulation of our research hypotheses and provides a comprehensive background for setting our research objectives. We appreciate your suggestion, as it guided us to enhance the depth and relevance of the introduction, thereby clearly articulating the significance of our study within the existing body of knowledge.

4. Clearly articulate why China is a particularly relevant or unique case for studying this paradox. This will help in establishing the significance of your research.

Response 4: In our revised manuscript, we have added a section in the introduction (fourth paragraph) to clearly articulate the unique relevance of China as a case study for exploring the Easterlin Paradox. The text now reads: 

“… China serves as a particularly compelling case for analyzing the Easterlin Paradox, marked by its unmatched economic growth and substantial socio-economic shifts over recent decades. Unlike the steady and gradual economic progress seen in many Western countries, China has witnessed rapid industrialization, urbanization, and economic development at an unprecedented rate and scale. This transformation is characterized not only by its temporal aspect but also by significant economic disparities and variations in income distribution across and within its provinces. Our research aims to explore how these swift economic changes and notable shifts in income distribution impact individuals' perceptions of happiness. Focusing on China, we intend to provide insights that critically assess and refine the understanding of the Easterlin Paradox globally, promoting a more detailed and context-sensitive analysis.”

This enhancement underscores the significance of our research by situating China as a critical case study. We appreciate your guidance, as it has enabled us to more effectively establish the importance and relevance of our study within the broader academic discourse.

5. You mention Hedonic Adaptation Theory and Relative Deprivation Theory but could integrate these more cohesively into your argument. Explain how these theories underpin your hypotheses.

Response 5: As highlighted in our response to the third point, we have refined the theoretical derivation in our revised manuscript to ensure a seamless integration with our hypotheses (Introduction, paragraphs 7-9). This revision includes a more cohesive discussion of both Hedonic Adaptation Theory and Relative Deprivation Theory, elucidating how these theories not only inform the foundation of our research but also directly underpin our hypotheses. We appreciate your suggestion, as it has significantly contributed to strengthening the theoretical grounding of our work and its connection to the proposed hypotheses.

6. Offer a brief preview of your methodology, especially the use of a machine learning model and panel data. This sets the stage for what makes your approach novel or robust.

Response 6: In response to your suggestion, we have enhanced the final paragraph of the introduction to provide a brief preview of our methodology, highlighting its innovative aspects. The revised text is as follows: 

“… A significant innovation in our methodology is the application of a machine-learning model to analyze large-scale panel data, specifically focusing on the dynamic interaction between economic growth, income inequality, and SWB. This approach is particularly novel due to its use of social media big data as a proxy for continuous SWB measurement across diverse regions and time frames. Traditional psychological methods, such as questionnaires, are impractical for the extensive sampling required across China's thirty-plus provinces over a decade. In contrast, social media platforms like Sina Weibo, with its 566 million monthly active users and a dataset extending back to 2010 (Tandoc & Eng, 2017; Weibo-Corporation, 2021), provide a rich source for non-invasive public psychology and behavior analysis through big data analytics of user activities (Barbier & Liu, 2011; Huang et al., 2022; Li et al., 2013). By leveraging this extensive dataset, our study employs a validated machine-learning model (Hao et al., 2015; Hao et al., 2014; Wang et al., 2020) to assess SWB based on the text of social media posts. This innovative approach allows us to explore the effects of economic growth and income inequality on SWB in both temporal and cross-sectional dimensions. Our findings aim to contribute a fresh perspective on the Easterlin paradox, particularly in the context of China’s unique socio-economic landscape.”

This revision succinctly outlines the novel and robust aspects of our methodology, setting the stage for the detailed explanation that follows in the methods section. We are grateful for your guidance, as it has allowed us to more effectively communicate the innovative nature of our research methodology.

7. consider providing a bit more context or rationale for each. This can help readers understand how you arrived at these specific hypotheses.

Response 7: As previously addressed in our responses to the third and fifth comments, we have enhanced paragraphs 7-9 of the introduction in our revised manuscript to address this point. This revision involves providing a more comprehensive context and rationale for our hypotheses, ensuring a seamless connection between the theoretical frameworks discussed earlier in the introduction and the specific hypotheses of our study. By elaborating on the theoretical underpinnings and the empirical observations that inform our hypotheses, we aim to offer readers a clearer understanding of the logical progression that led to the formulation of these hypotheses. We are grateful for your recommendation, as it has significantly contributed to enhancing the clarity and persuasiveness of our manuscript.

8. Tie your study to contemporary societal or policy issues, such as the ongoing debates about economic growth and well-being. This can enhance the real-world relevance of your research.

Response 8: In accordance with your suggestion to link our study more closely to contemporary societal and policy issues, we have expanded the discussion in the second paragraph of the revised introduction to cover these aspects. The new text reads as follows: 

"… Nonetheless, the association between economic growth and SWB has not gone unchallenged, especially when considering contemporary societal and policy issues such as debates around economic growth, environmental sustainability, and well-being. Evidence from several studies suggests that SWB correlates significantly with the material standard of living, highlighting the importance of integrating these findings into policy discussions to address the multifaceted dimensions of well-being beyond mere economic indicators (Gariepy et al., 2017; Sacks et al., 2010). Stevenson and Wolfers contended that there is no empirical evidence to suggest that a country's SWB plateaus after reaching a certain economic threshold, a point that gains relevance in the context of ongoing debates about the sustainability of economic growth and its implications for societal well-being (Stevenson & Wolfers, 2008). Broadly speaking, Easterlin's findings enjoy robust support in many developed nations such as Western Europe, Japan, and Korea, serving as a critical reference point for policymakers aiming to balance economic growth with the enhancement of societal well-being (Clark et al., 2008; Easterlin, 1995; Xing, 2011). However, results from some developing countries exhibit greater variability, suggesting that the path to well-being through economic growth may differ significantly across contexts, further underscoring the need for nuanced policy approaches that consider local socio-economic dynamics (Bortolotti & Biggeri, 2022; Clark, 2016; Opfinger, 2015; Stevenson & Wolfers, 2008)."

This revision is designed to underscore the real-world relevance of our research by situating it within ongoing debates on the sustainability of economic growth, environmental concerns, and their implications for well-being. By doing so, we aim to highlight the critical importance of our findings for informing policy discussions and crafting strategies that balance economic development with the holistic enhancement of societal well-being. We appreciate your guidance, as it has significantly enriched the contextual framing of our study, making it more relevant and impactful.

Methods

1. Ensure that all technical terms, especially those specific to economics and machine learning, are clearly defined or explained. This aids in accessibility for readers who may not be familiar with these fields.

Response 1: In response to your valuable suggestion, we have meticulously reviewed our manuscript to ensure that all technical terms and jargon, particularly those pertaining to economics and machine learning, such as Application Programming Interface (API), TextMind, and the Gini coefficient, are accompanied by straightforward explanations or appropriately cited references. This careful scrutiny and subsequent clarification aim to enhance the accessibility of our research to readers who may not possess specialized knowledge in these fields. We are grateful for your guidance, as it has significantly contributed to improving the clarity of our manuscript.

2. adding a bit more detail on how you addressed potential biases in social media data could be beneficial. This might include discussing the representativeness of the Sina Weibo user base or the limitations of social media data.

Response 2: In light of your suggestion, we have expanded our manuscript to more thoroughly address potential biases inherent in social media data. This expansion is reflected in both the methods and discussion sections of our revised manuscript. The relevant additions are as follows:

In the methods section, we described our data collection and preprocessing steps in detail (“Data Collection and Preprocessing,” first and second paragraph): 

“… text data was sourced from an initial pool of over 1.16 million Sina Weibo users (Li et al., 2013)…” “… Following established data collection methods (Huang et al., 2020; Huang et al., 2022; Li et al., 2020; Wang et al., 2020), we first retrieved public posts from 1.16 million mainland Chinese users via Weibo's API. Active users were then identified based on criteria including: (a) registration before January 1, 2010; (b) the exclusion of institutional, commercial, or celebrity accounts; and (c) a minimum of 500 original posts during the observation period. These criteria ensured the analysis focused on everyday shares from the general public, avoiding political or commercial propaganda, and aimed to include as broad a user base as possible. This led to the selection of 644,243 active Weibo users from thirty-one provinces for this study.”

Furthermore, in the discussion section, we acknowledged limitations related to our data source (“4 Discussion,” seventh paragraph):

 “… Firstly, our reliance on social media for data collection may exclude certain populations, such as individuals who are not literate or do not use social media. Over 78% of our sample from Sina Weibo did not disclose age data, constraining our ability to assess the impact of age demographics (Pavlova, 2021). Future studies should employ varied sampling techniques to address this limitation…”

These additions aim to transparently discuss the representativeness of the Sina Weibo user base and the limitations associated with social media data, thereby enhancing the rigor and credibility of our research. We appreciate your feedback, as it has guided us in refining our methodology to better account for and articulate potential biases and limitations of our study.

3. You’ve detailed how SWB and other variables were calculated. Explain why these particular variables were chosen and how they are expected to contribute to your research objectives would provide more context.

Response 3: In our revised manuscript, we have elaborated on the theoretical and empirical justifications for the selection of SWB indicators and related economic variables, thereby providing a more comprehensive context for their inclusion in our study. Key excerpts from the revised sections include:

Regarding the choice of SWB indicators: 

“… This study conceptualizes SWB through Ryff's six-factor model (Ryff, 1989), evaluating individuals' happiness by analyzing six psychological dimensions: self-acceptance, positive relationships with others, autonomy, environmental mastery, sense of purpose and meaning in life, and personal growth. This theoretical basis and measurement method receive high praise in the positive psychology arena (Hao et al., 2015; Hao et al., 2013; Ryff, 1989)…”

On the selection of economic indicators: 

“…Aligned with the methodologies utilized in existing research (Diener & Tov, 2007; Yang et al., 2020), per capita Gross Domestic Product (PRGDP) is adopted as a metric for assessing the absolute economic growth of each province. PRGDP is defined as the quotient of the total GDP value and the regional population for a specified year. By adjusting for variations in city sizes, PRGDP provides a more accurate depiction of regional economic conditions compared to GDP alone (Huang et al., 2019; Huang et al., 2022)…”

And on the rationale behind using the Gini coefficient for income inequality: 

“… In accordance with established economic research paradigms, the Gini coefficient (Gi), first introduced by Corrado Gini (Gini, 1921; Tian, 2012), is utilized as the indicator to measure income inequality within each province (Sundrum, 2003). This coefficient theoretically ranges from 0, indicating complete income equality, to 1, denoting absolute income inequality. Elevated values of the coefficient indicate higher levels of income disparity within a country or region…”

These additions aim to clarify the rationale behind the selection of specific variables and their expected contributions to our research objectives. By detailing the theoretical and empirical underpinnings of our methodological choices, we seek to provide readers with a clearer understanding of how these variables are pivotal in exploring the dynamics between economic factors and subjective well-being. We are grateful for your feedback, as it has significantly enhanced the depth and clarity of our methodology section.

4. Elaborate on why this approach was chosen, its benefits, and any limitations it might have compared to traditional methods.

Response 4: Indeed, we acknowledge that the rationale behind our methodological approach is intrinsically linked to our research hypotheses and is thus more appropriately addressed within the introduction, as we have implemented in response to your sixth point regarding the introduction. As mentioned, we elucidated this matter in the last paragraph of the introduction, discussing the selection of our approach and its advantages in comparison to more traditional methodologies. Concurrently, in the methods section under “Statistical Analysis,” we provide a detailed account of how these methods were specifically implemented in our study. We appreciate your guidance, as it has been instrumental in refining our manuscript to better articulate the justification and implementation of our research methodology.

5. If there are any ethical considerations, especially related to the use of social media data, they should be explicitly stated. This might include how user privacy was protected or how data anonymization was handled.

Response 5: Following your recommendation and in line with editor's requirements, we have added an Ethics Statement at the end of the methods section to address ethical considerations related to the use of social media data. The newly included statement reads as follows:

“The research initiative received approval from the Ethics Committee, Institute of Psychology, Chinese Academy of Sciences (project number: H15009). All posts analyzed in this study were sourced from Sina Weibo, with stringent measures taken to preserve user privacy. The identities, usernames, and original content of the users' posts were deliberately omitted from the analysis, focusing exclusively on the examination of data at the provincial level. Given that Sina Weibo is publicly accessible and in adherence to the established protocols and ethical standards within the research domain (Kosinski et al.), the necessity for informed consent was exempted by the Ethics Committee, Institute of Psychology, Chinese Academy of Sciences.”(“Ethics Statement”)

This addition ensures that our manuscript explicitly addresses how user privacy was protected and how data anonymization was handled, thereby underscoring our commitment to ethical research practices. We appreciate your guidance, as it has significantly contributed to enhancing the transparency and ethical considerations of our study.

Results

1. consider simplifying the presentation for better readability. Use clear subheadings to separate different aspects of the results, such as demographics, descriptive statistics, main effects estimation, and interaction analysis.

Response 1: In response to your suggestion, we have ensured that the revised manuscript adopts a structured approach to presenting the results, utilizing clear subheadings to differentiate between various aspects of the findings. We appreciate your guidance, as it has been instrumental in enhancing the presentation and accessibility of our results.

2. provide brief interpretations of what these statistics mean in terms of your research question. Discuss the practical implications of these findings, especially in the context of policy and economic planning.

Response 2: Thank you for your comment. However, in keeping with the conventional reporting standards aimed at maintaining conciseness and readability within the "Results" section, we have adhered to a straightforward presentation of statistical analysis outcomes. This approach ensures that the results are presented in a clear, unambiguous manner, allowing for direct interpretation in relation to our research question.

Furthermore, we have dedicated a significant portion of the "Discussion" section to exploring the practical implications of our findings, particularly in terms of their relevance to policy and economic planning.

3. Elaborate on the implications of the findings related to the Gini coefficient and PRGDP. How do these results contribute to existing literature, and what do they suggest for future economic and social policies?

Response 3: Thank you for your comment. As mentioned in response to the second point, we prefer to elaborate on this content in the "Discussion" section.

Discussion

1. You've adeptly connected your findings to hedonic adaptation theory and relative deprivation theory. Further elaborating on how your results extend or challenge these theories could add depth to your discussion.

Response 1: In our revised manuscript, we have further explored the connections between our findings and two pivotal theories—hedonic adaptation theory and relative deprivation theory—within the "Discussion" section, specifically in the second and third paragraphs, respectively.Regarding the hedonic adaptation theory, we detailed: 

“… However, when employing a more flexible model, we find a statistically significant positive correlation between the two variables across both time and cross-sectional dimensions, lending empirical support to our research hypothesis H1. These findings are in concordance with the hedonic adaptation theory's explanation of the Easterlin paradox, suggesting that the influence of wealth accumulation on happiness is not linear but is influenced by its baseline levels (Easterlin & Angelescu, 2009; Knight & Gunatilaka, 2011). It is crucial to clarify that the observed elasticity in the relationship between economic growth and subjective well-being does not negate the claims of the Easterlin paradox. Rather, it indicates that as a region experiences exponential economic growth, there is a corresponding increase in the population's subjective well-being. Specifically, for every 46.70% increase in GDP per capita in the time series, subjective well-being rises by 0.38; in the cross-sectional analysis, a similar increase in GDP per capita yields a 0.51 rise in subjective well-being. Thus, economic factors remain pertinent, but the demand and expectation for growth have intensified.”

Furthermore, concerning the relative deprivation theory, we elaborated: 

“… Income inequality exerts a deleterious impact on subjective well-being, corroborating the tenets of relative deprivation theory (Davis, 1959; Walker & Pettigrew, 1984). In our study, for each 0.09 unit increase in income inequality in the time-series analysis, we observe a corresponding 1.47 unit decrease in subjective well-being, thereby confirming our Hypothesis H2 and aligning with extant literature. Luttmer posits that income inequality within a community skews the local mean income to the right of the median, representing a wealthier minority. Consequently, the majority of the population earns below the local mean, resulting in a decline in their 'relative income' (Luttmer, 2005). Furthermore, our research illuminates a previously overlooked aspect: the interaction between income inequality and economic growth vis-à-vis subjective well-being. Specifically, the positive predictive influence of economic growth on subjective well-being diminishes as income inequality escalates. It is anticipated that the enhancement in well-being owing to economic growth vanishes entirely when the Gini coefficient attains a value of 0.61. This substantiates our Hypothesis H3. In accordance with relative deprivation theory, we surmise that the crux of this interaction lies in the widening wealth gap, which intensifies social comparisons and predominantly instills a sense of loss in the general populace (Liu et al., 2019; Panning, 2014).”

These detailed discussions aim to provide a deeper understanding of how our findings both align with and expand upon existing theories. We are grateful for your suggestion, as it has significantly enriched the discussion of our study's implications and theoretical contributions.

2. provide a clearer articulation of how these insights can inform policy-making or economic planning would be beneficial.

Response 2: In the "Discussion" section's fifth paragraph, we have added content to more clearly articulate how our insights can inform policy-making and economic planning. The text reads as follows:

"… These findings highlight the complex relationship between economic growth, income inequality, and subjective well-being, offering direction for policy development. To improve societal well-being, policies must foster sustainable economic growth and introduce initiatives to reduce income disparity. Recommended strategies encompass progressive taxation, enhancements in social welfare, policies promoting equal opportunities, and strengthening oversight mechanisms to deter tax evasion by affluent sectors and individuals. Such an approach to narrowing the wealth gap ensures that economic advancements lead to significant societal benefits. Thus, our study advocates for an integrated strategy in economic planning that not only seeks inclusive growth but also aims to boost overall happiness by ensuring equitable contributions across all economic sectors."

This addition aims to provide a clearer understanding of how the interplay between economic factors and subjective well-being can guide the formulation of policies aimed at achieving a more equitable and happy society. We appreciate your suggestion, as it has significantly enhanced the practical relevance and applicability of our research findings.

3. Delv deeper into how your findings align with or diverge from these studies could offer a richer context.

Response 3: As mentioned in the response to the second point, in the revised manuscript, we have delved deeper into how our findings both align with and diverge from previous studies in paragraphs 2 and 3 of the "Discussion" section. This examination provides a richer context by comparing our study's outcomes with the established literature, particularly regarding the Easterlin paradox and theories of hedonic adaptation and relative deprivation. 

Overall, our analysis reveals that while our findings initially seem to support the Easterlin paradox through the observed linear relationship between economic growth and subjective well-being in China, a more nuanced investigation using a flexible model uncovers a significant positive correlation across both time and cross-sectional dimensions. This nuanced finding challenges and extends the traditional interpretation of the Easterlin paradox by demonstrating that the impact of wealth accumulation on happiness is moderated by baseline levels of well-being, thus indicating a complex and non-linear relationship. Furthermore, our study confirms the adverse effects of income inequality on subjective well-being, in line with relative deprivation theory, and introduces new insights into the interaction between economic growth and income inequality. This aspect of our findings highlights the importance of considering both economic growth and income distribution in discussions of societal well-being and policy formulation. By critically comparing our results with existing research, we contribute to a deeper understanding of the intricate dynamics between economic factors and subjective well-being. 

We appreciate your suggestion, as it has significantly enriched the discussion section by providing a more comprehensive examination of how our research extends and challenges the current body of knowledge.

4. Expanding on how limitations might impact the interpretation of your results and suggesting ways future research could overcome them would strengthen your discussion.

Response 4: In our revised manuscript, we have expanded the section addressing the study's limitations to ensure a comprehensive understanding of the scope and applicability of our findings. The text now includes the following considerations:

"… Firstly, our reliance on social media for data collection may exclude certain populations, such as individuals who are not literate or do not use social media. Over 78% of our sample from Sina Weibo did not disclose age data, constraining our ability to assess the impact of age demographics (Pavlova, 2021). Future studies should employ varied sampling techniques to address this limitation. Secondly, as this study concentrates on the macro-social link between economic variables and collective subjective well-being, the findings might not be directly applicable to individual experiences, highlighting the potential for ecological fallacies. Future research could engage in individual-level investigations through surveys and experimental methods to draw comparisons with macro-level data. Thirdly, while our analysis focuses solely on the impact of objective economic indicators on subjective well-being, it does not delve into the psychological mechanisms underlying this relationship. Further studies are encouraged to explore mediating variables such as psychological expectations (Niessen et al., 2023), social support (Hou et al., 2022), and cultural participation (Giovanis et al., 2021), in line with theories of hedonic adaptation and relative deprivation…"

This elaboration aims to transparently outline the limitations of our research approach and suggest directions for future studies to build upon our work. We appreciate your guidance, as it has significantly contributed to strengthening the discussion of our study's limitations and potential areas for future research.

5. Discuss potential applications of your findings beyond the Chinese context. How might these insights apply to other economies or inform global economic policies?

Response 5: In line with your suggestion and similar to our approach in addressing the fourth point, we have included reflections on the broader applicability of our findings in the "Discussion" section, particularly in the seventh paragraph, to consider potential applications beyond the Chinese context. The revised text is as follows:

"… while our analysis focuses solely on the impact of objective economic indicators on subjective well-being, it does not delve into the psychological mechanisms underlying this relationship. Further studies are encouraged to explore mediating variables such as psychological expectations (Niessen et al., 2023), social support (Hou et al., 2022), and cultural participation (Giovanis et al., 2021) … it is hypothesized that socio-cultural factors, including the emphasis on collective well-being, Confucian ethics' influence on notions of fairness and prosperity, and intra-national cultural differences, could significantly affect the interplay between economic growth, income inequality, and subjective well-being. An in-depth examination of these elements might offer a more detailed perspective on how economic and social policies impact happiness in the Chinese context…"

While our data and analysis are specifically rooted in the Chinese socio-economic and cultural milieu, the underlying principles and findings may offer valuable insights for other economies, especially those undergoing rapid growth and facing challenges related to income inequality and social welfare. We appreciate your suggestion, as it has encouraged us to consider the broader implications of our research and its relevance to a global audience.

6. Reflect on the strengths and limitations of using social media data and machine learning methodologies in your research. How do these methods compare to traditional survey-based approaches in studying subjective well-being?

Response 6: Thank you for your suggestion. This issue relates to the points raised in the introduction (the sixth comment) and in the methods section (the fourth comment) regarding our choice of methodology. Specifically, we have elucidated our rationale for employing social media data and machine learning methodologies in the introduction, explaining why these approaches were selected for our study. Additionally, the limitations associated with using machine learning to study subjective well-being (SWB) have been addressed in the discussion of limitations and future directions mentioned in the fifth comment.

Using social media data and machine learning methodologies offers distinct strengths, such as the ability to analyze large-scale datasets in real-time and capture a wide range of subjective well-being expressions across diverse demographic groups. This approach allows for a nuanced examination of SWB in a naturalistic setting, providing insights that might not be easily accessible through traditional survey methods. However, these methods also present limitations, including potential biases in social media usage across different populations. In comparison to traditional survey-based approaches, our methodology offers a complementary perspective by leveraging big data. While surveys provide direct, self-reported measures of well-being, social media data and machine learning can analyze spontaneous expressions of well-being at scale. Each approach has its unique advantages and limitations, suggesting that a combined methodology could offer the most comprehensive insights into subjective well-being.

We appreciate your guidance, as it has prompted us to reflect on the strengths and limitations of our chosen methodologies, enhancing our discussion of how these methods contribute to the broader field of SWB research.

7. Suggest specific areas for future research that could build on your findings. For instance, exploring individual-level data or integrating psychological variables could provide further insights.

Response 7: Thank you for your suggestion. We have integrated this content into the limitations and future directions section of the "Discussion" part in our revised manuscript. As this study concentrates on the macro-social link between economic variables and collective subjective well-being, the findings might not be directly applicable to individual experiences, highlighting the potential for ecological fallacies. Future research could engage in individual-level investigations through surveys and experimental methods to draw comparisons with macro-level data. We appreciate your guidance, as it has significantly contributed to refining our manuscript by identifying clear directions for future research that can build on our study's findings, thereby advancing the field of SWB research.

8. on your findings, offer specific recommendations for economic policies that could enhance subjective well-being while addressing income inequality.

Response 8: As highlighted in response to the second suggestion, and to address your recommendation, we have incorporated the following content into the discussion of our revised manuscript:

"... These findings underscore the complex interplay between economic growth, income inequality, and subjective well-being, providing valuable guidance for policy formulation. To enhance societal well-being, it is imperative for policies to support sustainable economic growth while implementing measures to mitigate income disparity. We recommend several strategies, including the adoption of progressive taxation, the enhancement of social welfare programs, the promotion of equal opportunities, and the strengthening of oversight mechanisms to prevent tax evasion by wealthy individuals and sectors. By adopting such measures to narrow the wealth gap, we can ensure that economic progress translates into widespread societal benefits. Therefore, our study calls for a holistic approach to economic planning, one that not only pursues inclusive growth but also actively works to increase overall happiness by promoting fair contributions across all segments of the economy."

We appreciate your suggestion, as it has enriched our discussion by offering clear, policy-oriented directions that stem from our research findings.

9. Discuss any socio-cultural factors that might influence the relationship between economic growth, income inequality, and subjective well-being, particularly in the Chinese context.

Response 9: In recognition of the unique cultural context of our dataset, we have included considerations of socio-cultural factors in the future directions section of our revised manuscript. The text now reads as follows:

"… it is hypothesized that socio-cultural factors, including the emphasis on collective well-being, Confucian ethics' influence on notions of fairness and prosperity, and intra-national cultural differences, could significantly affect the interplay between economic growth, income inequality, and subjective well-being. An in-depth examination of these elements might offer a more detailed perspective on how economic and social policies impact happiness in the Chinese context."

This addition acknowledges the potential impact of socio-cultural factors on the relationship between economic growth, income inequality, and subjective well-being, particularly within the Chinese context. We appreciate your suggestion, as it has significantly enriched our discussion by emphasizing the need for a culturally sensitive understanding of well-being in economic research.

10. End your discussion with strong concluding remarks that summarize the key takeaways of your study and their broader significance.

Response 10: In response to your valuable feedback, we have included an additional paragraph at the end of the "Discussion" section in our revised manuscript. This paragraph synthesizes the principal findings and broader implications of our research, stating:

“In conclusion, our study enriches the discourse on the intricate dynamics between economic growth, income inequality, and subjective well-being through an innovative analysis that combines macroeconomic indicators with social media analytics. By revealing the bidirectional influences where economic growth positively correlates with subjective well-being and income inequality negatively affects it, we offer nuanced insights into the debate surrounding the Easterlin paradox within the context of China's rapid economic evolution. Our findings challenge the traditional understanding of the paradox, demonstrating that economic growth can indeed enhance subjective well-being, provided income inequality is addressed effectively. This underscores the critical need for integrated economic policies that promote not only growth but also equity, to ensure that the benefits of economic advancements are widely shared across the society. Moreover, our methodological approach, leveraging social media data, opens new avenues for real-time, nuanced analysis of subjective well-being, offering a template for future research in the field. As we navigate the complexities of economic development and social welfare, the importance of fostering policies that minimize income disparities while bolstering economic growth has never been more evident. Our study ultimately advocates for a balanced approach to economic planning, one that equally prioritizes the pursuit of happiness and the equitable distribution of wealth, reflecting David Hume's assertion that the ultimate goal of human industry is the attainment of happiness.”

We hope this addition adequately addresses your concern by concisely summarizing the key insights of our research and highlighting their significant implications for both policy and future studies. Thank you for your constructive comment, which has undoubtedly enhanced the quality and impact of our manuscript.

We appreciate for the Editor and Reviewers’ warm work earnestly. Thank you very much for your comments and suggestions!

References:

Barbier, G., & Liu, H. (2011). Data Mining in Social Media. In C. C. Aggarwal (Ed.), Social Network Data Analytics (pp. 327-352). Springer US. https://doi.org/10.1007/978-1-4419-8462-3_12

Bortolotti, L., & Biggeri, M. (2022). Is the slowdown of China's economic growth affecting multidimensional well-being dynamics? Structural Change and Economic Dynamics, 63, 478-489. https://doi.org/10.1016/j.strueco.2022.07.003

Clark, A. E. (2016). Adaptation and the Easterlin Paradox. In T. Tachibanaki (Ed.), Advances in Happiness Research (pp. 75-94). Springer Japan. https://doi.org/10.1007/978-4-431-55753-1_6

Clark, A. E., Frijters, P., & Shields, M. A. (2008). Relative Income, Happiness, and Utility: An Explanation for the Easterlin Paradox and Other Puzzles. Journal of Economic Literature, 46(1), 95-144. https://doi.org/10.1257/jel.46.1.95

Davis, J. A. (1959). A Formal Interpretation of the Theory of Relative Deprivation. SOCIOMETRY, 22(4), 280-296. https://doi.org/10.2307/2786046

Diener, E., & Tov, W. (2007). Subjective Well-Being and Peace. Journal of Social Issues, 63(2), 421-440. https://doi.org/10.1111/j.1540-4560.2007.00517.x

Easterlin, R. A. (1995). Will raising the incomes of all increase the happiness of all? Journal of Economic Behavior & Organization, 27(1), 35-47. https://doi.org/10.1016/0167-2681(95)00003-b 

Easterlin, R. A., & Angelescu, L. (2009). Happiness and Growth the World Over: Time Series Evidence on the Happiness-Income Paradox. SSRN Electronic Journal. https://doi.org/10.2139/ssrn.1369806

Gao, R., Hao, B., Li, H., Gao, Y., & Zhu, T. (2013). Developing simplified Chinese psychological linguistic analysis dictionary for microblog. International conference on brain and health informatics, 

Gariepy, G., Elgar, F. J., Sentenac, M., & Barrington-Leigh, C. (2017). Early-life family income and subjective well-being in adolescents [Article]. PLoS One, 12(7), Article e0179380. https://doi.org/10.1371/journal.pone.0179380

Gini, C. (1921). Measurement of Inequality of Incomes. The Economic Journal, 31(121), 124-126. https://doi.org/10.2307/2223319

Giovanis, E., Akdede, S. H., & Ozdamar, O. (2021). Impact of the EU Blue Card programme on cultural participation and subjective well-being of migrants in Germany. PLoS One, 16(7), Article e0253952. https://doi.org/10.1371/journal.pone.0253952

Hao, B., Li, A., Bai, S., & Zhu, T. (2015). Predicting psychological features based on web behavioral data: Mental health status and subjective well-being. Chinese Science Bulletin, 60(11), 994-1001. https://doi.org/10.1360/n972014-00763

Hao, B., Li, L., Gao, R., Li, A., & Zhu, T. (2014). Sensing Subjective Well-Being from Social Media. In D. Slezak, G. Schaefer, S. T. Vuong, & Y. S. Kim (Eds.), Active Media Technology (Vol. 8610, pp. 324-335). https://doi.org/10.1007/978-3-319-09912-5_27

Hao, B., Li, L., Li, A., & Zhu, T. (2013). Predicting Mental Health Status on Social Media. In Cross-Cultural Design. Cultural Differences in Everyday Life (pp. 101-110). Springer Berlin Heidelberg. https://doi.org/10.1007/978-3-642-39137-8_12

Hou, T., Zhang, R., Xie, Y., Yin, Q., Cai, W., Dong, W., & Deng, G. (2022). Education and subjective well-being in Chinese rural population: A multi-group structural equation model [Article]. PLoS One, 17(3), Article e0264108. https://doi.org/10.1371/journal.pone.0264108

Huang, F., Ding, H., Liu, Z., Wu, P., Zhu, M., Li, A., & Zhu, T. (2020). How fear and collectivism influence public's preventive intention towards COVID-19 infection: a study based on big data from the social media. BMC Public Health, 20(1), 1707. https://doi.org/10.1186/s12889-020-09674-6

Huang, F., Li, H., Ding, H., Wu, S., Liu, M., Liu, T., Liu, X., & Zhu, T. (2019). Influence model of economic development on collective morality from the perspective of social media big data. Chinese Science Bulletin, 65(19), 2062-2070. https://doi.org/10.1360/tb-2019-0702

Huang, F., Li, S., Ding, H., Han, N., & Zhu, T. (2022). Does more moral equal less corruption? The different mediation of moral foundations between economic growth and corruption in China. Current Psychology. https://doi.org/10.1007/s12144-022-03735-2

Knight, J., & Gunatilaka, R. (2011). Does Economic Growth Raise Happiness in China? Oxford Development Studies, 39(1), 1-24. https://doi.org/10.1080/13600818.2010.551006

Kosinski, M., Matz, S. C., Gosling, S. D., Popov, V., & Stillwell, D. Facebook as a research tool for the social sciences: Opportunities, challenges, ethical considerations, and practical guidelines. American Psychologist, 70(6), 543-556. https://doi.org/10.1037/a0039210

Li, A., Zhang, F., & Zhu, T. (2013). Web Use Behaviors for Identifying Mental Health Status. In K. Imamura, S. Usui, T. Shirao, T. Kasamatsu, L. Schwabe, & N. Zhong (Eds.), Brain and Health Informatics (Vol. 8211, pp. 348-358). https://doi.org/10.1007/978-3-319-02753-1_35

Li, S., Wang, Y., Xue, J., Zhao, N., & Zhu, T. (2020). The Impact of COVID-19 Epidemic Declaration on Psychological Consequences: A Study on Active Weibo Users. Int J Environ Res Public Health, 17(6), Article 2032. https://doi.org/10.3390/ijerph17062032

Liu, Y., Zhang, F., Liu, Y., Li, Z., & Wu, F. (2019). Economic disadvantages and migrants' subjective well-being in China: The mediating effects of relative deprivation and neighbourhood deprivation. Population, Space and Place, 25(2), e2173. https://doi.org/10.1002/psp.2173

Luttmer, E. F. P. (2005). Neighbors as Negatives: Relative Earnings and Well-Being. The Quarterly Journal of Economics, 120(3), 963-1002. https://doi.org/10.1093/qje/120.3.963

Niessen, D., Wicht, A., & Lechner, C. M. (2023). Aspiration-attainment gaps predict adolescents' subjective well-being after transition to vocational education and training in Germany [Article]. PLoS One, 18(6), Article e0287064. https://doi.org/10.1371/journal.pone.0287064

Opfinger, M. (2015). The Easterlin paradox worldwide. Applied Economics Letters, 23(2), 85-88. https://doi.org/10.1080/13504851.2015.1051650

Panning, W. H. (2014). Inequality, Social Comparison, and Relative Deprivation. American Political Science Review, 77(2), 323-329. https://doi.org/10.2307/1958918

Pavlova, M. K. (2021). Do workers accumulate resources during continuous employment and lose them during unemployment, and what does that mean for their subjective well-being? [Article]. PLoS One, 16(12), Article e0261794. https://doi.org/10.1371/journal.pone.0261794

Ryff, C. D. (1989). Beyond ponce de leon and life satisfaction: New directions in quest of successful ageing. International Journal of Behavioral Development, 12(1), 35-55. https://doi.org/10.1177/016502548901200102

Sacks, D., Stevenson, B., & Wolfers, J. (2010). Subjective Well-Being, Income, Economic Development and Growth. National Bureau of Economic Research Working Paper Series, No. 16441. https://doi.org/10.3386/w16441

Stevenson, B., & Wolfers, J. (2008). Economic Growth and Subjective Well-Being: Reassessing the Easterlin Paradox [Article; Proceedings Paper]. Brookings Papers on Economic Activity, 39(1), 1-28. https://doi.org/10.3386/w14282

Sundrum, R. M. (2003). Income Distribution in Less Developed Countries (1st ed.). Routledge. https://doi.org/10.4324/9780203168493

Tandoc, E. C., & Eng, N. (2017). Climate Change Communication on Facebook, Twitter, Sina Weibo, and Other Social Media Platforms. Oxford Research Encyclopedia of Climate Science. https://doi.org/10.1093/acrefore/9780190228620.013.361

Tausczik, Y. R., & Pennebaker, J. W. (2009). The Psychological Meaning of Words: LIWC and Computerized Text Analysis Methods. Journal of Language and Social Psychology, 29(1), 24-54. https://doi.org/10.1177/0261927x09351676

Tian, W. (2012). Calculation of Gini coefficient of provincial residents' income and analysis of its changing trend. Economic Science(02), 48-59. https://doi.org/10.19523/j.jjkx.2012.02.004

Walker, I., & Pettigrew, T. F. (1984). Relative deprivation theory: An overview and conceptual critique. British Journal of Social Psychology, 23(4), 301-310. https://doi.org/10.1111/j.2044-8309.1984.tb00645.x

Wang, Y., Wu, P., Liu, X., Li, S., Zhu, T., & Zhao, N. (2020). Subjective Well-Being of Chinese Sina Weibo Users in Residential Lockdown During the COVID-19 Pandemic: Machine Learning Analysis. J Med Internet Res, 22(12), e24775. https://doi.org/10.2196/24775

Weibo-Corporation. (2021). Weibo Reports Second Quarter 2021 Unaudited Financial Results. PR Newswire. https://www.prnewswire.com/news/weibo-corporation/

Xing, Z. (2011). A study of the relationship between income and subjective well-being in China. Sociological Studies, 25(01), 196-219+245-246. https://doi.org/10.19934/j.cnki.shxyj.2011.01.008

Yang, X., Geng, L., & Zhou, K. (2020). Environmental pollution, income growth, and subjective well-being: regional and individual evidence from China. Environ Sci Pollut Res Int, 27(27), 34211-34222. https://doi.org/10.1007/s11356-020-09678-0

Zhao, N., Jiao, D., Bai, S., & Zhu, T. (2016). Evaluating the Validity of Simplified Chinese Version of LIWC in Detecting Psychological Expressions in Short Texts on Social Network Services. PLoS One, 11(6), e0157947. https://doi.org/10.1371/journal.pone.0157947

---

## [Editor Report · Decision Letter 1]

13 Mar 2024

Does Wealth Equate to Happiness? An 11-Year Panel Data Analysis Exploring Socio-Economic Indicators and Social Media Metrics

PONE-D-23-36509R1<o:p></o:p>

<o:p> </o:p>

Dear Dr. Zhu,<o:p></o:p>

We’re pleased to inform you that your manuscript has been judged scientifically suitable for publication and will be formally accepted for publication once it meets all outstanding technical requirements.<o:p></o:p>

Within one week, you’ll receive an e-mail detailing the required amendments. When these have been addressed, you’ll receive a formal acceptance letter and your manuscript will be scheduled for publication.<o:p></o:p>

An invoice will be generated when your article is formally accepted. Please note, if your institution has a publishing partnership with PLOS and your article meets the relevant criteria, all or part of your publication costs will be covered. Please make sure your user information is up-to-date by logging into Editorial Manager at http://www.editorialmanager.com/pone/ and clicking the ‘Update My Information' link at the top of the page. If you have any questions relating to publication charges, please contact our Author Billing department directly at authorbilling@plos.org.<o:p></o:p>

If your institution or institutions have a press office, please notify them about your upcoming paper to help maximize its impact. If they’ll be preparing press materials, please inform our press team as soon as possible -- no later than 48 hours after receiving the formal acceptance. Your manuscript will remain under strict press embargo until 2 pm Eastern Time on the date of publication. For more information, please contact onepress@plos.org.<o:p></o:p>

<o:p> </o:p>

Kind regards,<o:p></o:p>

Roghieh Nooripour, Ph.D

Academic Editor

PLOS ONE<o:p></o:p>

---

## [Editor Report · Acceptance letter]

18 Mar 2024

PONE-D-23-36509R1 

PLOS ONE

Dear Dr. Zhu, 

I'm pleased to inform you that your manuscript has been deemed suitable for publication in PLOS ONE. Congratulations! Your manuscript is now being handed over to our production team.

Kind regards, 

on behalf of

Dr. Roghieh Nooripour 

Academic Editor

PLOS ONE